# Inhibition of aquaporin-3 in macrophages by a monoclonal antibody as potential therapy for liver injury

Mariko Hara-Chikuma [1✉], Manami Tanaka[1,2], Alan S. Verkman[3] & Masato Yasui[1,2]

Aquaporin 3 (AQP3) is a transporter of water, glycerol and hydrogen peroxide ($H_2O_2$) that is expressed in various epithelial cells and in macrophages. Here, we developed an anti-AQP3 monoclonal antibody (mAb) that inhibited AQP3-facilitated $H_2O_2$ and glycerol transport, and prevented liver injury in experimental animal models. Using AQP3 knockout mice in a model of liver injury and fibrosis produced by $CCl_4$, we obtained evidence for involvement of AQP3 expression in nuclear factor-κB (NF-κB) cell signaling, hepatic oxidative stress and inflammation in macrophages during liver injury. The activated macrophages caused stellate cell activation, leading to liver injury, by a mechanism involving AQP3-mediated $H_2O_2$ transport. Administration of an anti-AQP3 mAb, which targeted an extracellular epitope on AQP3, prevented liver injury by inhibition of AQP3-mediated $H_2O_2$ transport and macrophage activation. These findings implicate the involvement of macrophage AQP3 in liver injury, and provide evidence for mAb inhibition of AQP3-mediated $H_2O_2$ transport as therapy for macrophage-dependent liver injury.

[1] Department of Pharmacology, School of Medicine, Keio University, Tokyo, Japan. [2] Keio Global Research Institute, Center for Water Biology and Medicine, Tokyo, Japan. [3] Departments of Medicine and Physiology, University of California San Francisco, San Francisco, CA, USA. ✉email: haramari@kuhp.kyoto-u.ac.jp

Chronic liver injury with inflammation and fibrosis occurs in various etiologies, including viral infection, autoimmune disorders, alcohol or drug abuse, metabolic disorders, and nonalcoholic steatohepatitis[1,2]. Liver fibrosis has been implicated in many types of liver diseases such as cirrhosis and hepatocellular carcinoma[1,3,4]. Although multiple cell populations in the liver contribute to various inflammatory and fibrogenic pathways, hepatic macrophages are considered a key contributor to acute and chronic liver inflammation and fibrosis[1,5–9]. Regardless of the underlying cause, it is now generally accepted that resident macrophages (Kupffer cells) and infiltrating macrophages are activated after acute liver cell death, which then exacerbates the initial liver injury by secretion of pro-inflammatory cytokines and chemokines that activate hepatic stellate cells (HSCs). Activated HSCs transdifferentiate into myofibroblasts, which amplify the progression of chronic liver fibrosis[1,2,7].

Previous studies have suggested the involvement of redox state and oxidative stress in the progression of liver fibrosis[10,11]. During liver injury, hepatocytes, neutrophils, and macrophages generate high levels of reactive oxygen species (ROS; $O_2^-$, OH, and hydrogen peroxide [$H_2O_2$]), which are thought to increase oxidative stress, cause direct cytotoxicity, or act as intracellular signaling mediators[10,11]. Pro-inflammatory macrophages show an increase in the inflammatory mediators nitric oxide and ROS, and enhanced aerobic glycolysis[12,13]. NADPH oxidase (Nox) and the mitochondrial respiratory pathway are the two major producers of endogenous ROS[14–17]. Previous studies have shown that Nox1, 2, and 4 in hepatocytes or HSCs are involved in liver inflammation and fibrosis via ROS production[18–20]. Thus, a variety of mechanisms for liver injury and fibrosis have been proposed though their relative importance remains unclear. Because advanced liver fibrosis is largely irreversible[1,3], there remains an unmet need for development of novel therapeutics to reduce liver inflammation and fibrosis in a variety of disorders including cirrhosis and hepatocellular carcinoma.

Aquaporin-3 (AQP3), a member of the aquaporin water channel family, functions as a transporter of water and small molecules, including glycerol and $H_2O_2$[21]. We previously reported that AQP3-mediated $H_2O_2$ transport and increased intracellular $H_2O_2$ concentration acted as a secondary messenger for cell signaling involving factors such as NF-κB and PTEN, which increased inflammation, cell proliferation, and cell migration[22–25]. AQP3 knockout (AQP3$^{-/-}$) mice showed reduced inflammation in models of contact hypersensitivity and psoriasis, and in cancer progression[22,23,25,26]. We speculated herein that AQP3 might regulate cellular ROS levels and thereby oxidative stress in oxidative stress-related diseases including liver fibrosis.

The present study examines the involvement and downstream mechanisms of AQP3-mediated $H_2O_2$ transport in liver injury in an experimental mouse model. An anti-AQP3 monoclonal antibody was generated that inhibited AQP3-mediated $H_2O_2$ transport and prevented acute and chronic liver injury.

## Results

### Reduced CCl$_4$-induced acute liver injury in AQP3$^{-/-}$ mice.
AQP3 expression has been reported in human liver macrophages by immunostaining[27]. Double immunofluorescence staining of normal human liver with anti-AQP3 and anti-CD68 (human macrophage marker) showed AQP3 expression predominantly in CD68$^+$ macrophages (Supplementary Fig. 1a). The AQP3 expression pattern was studied in mouse liver tissue. AQP3 immunostaining with anti-F4/80 (macrophage marker) and anti-desmin (HSC marker) showed AQP3 expression in macrophages and HSCs (Fig. 1a). RT-PCR in isolated hepatic cells from naive

wild-type (WT) mouse liver showed high AQP3 expression in hepatic macrophages and HSCs, and little AQP3 expression in hepatocytes (Fig. 1b, Supplementary Fig. 1b).

A CCl$_4$ model of acute liver injury was done in WT and AQP3$^{-/-}$ mice to investigate the potential role of AQP3 in the pathogenesis of liver injury[28,29]. Previous studies have shown macrophages play an important role in this model via the secretion of chemokines (e.g., CCL2 and CCL5) and inflammatory cytokines (e.g., tumor necrosis factor-α [TNF-α] and IL-1β) that promote HSC activation and transdifferentiation, and induce α-smooth muscle actin (α-SMA) and collagen α1[1,5]. CCl$_4$ injection greatly increased expression of mRNA encoding TNF-α, CCL2, and α-SMA in WT liver, which were much reduced in AQP3$^{-/-}$ liver (Fig. 1c). AQP3$^{-/-}$ mice also showed a significant reduction in the elevations in serum aspartate aminotransferase (AST) and alanine aminotransferase (ALT), the common biomarkers of liver injury[30] (Supplementary Fig. 1c). The numbers of CD11B$^{high}$ F4/80$^{int}$-positive recruited macrophages increased after CCl$_4$ injection, as described previously[28,29,31], but comparable in WT and AQP3$^{-/-}$ livers (Fig. 1d, Supplementary Fig. 1d), indicating similar macrophage infiltration into the liver.

We next determined whether AQP3 expression affects cellular ROS levels and oxidative stress during acute liver injury. Hepatocytes were isolated from liver 1 day after CCl$_4$ injection, and cellular ROS levels were assayed using the fluorescent dye CM-H$_2$DCFDA that reacts with ROS including $H_2O_2$[22,23,32]. Figure 1e shows that CCl$_4$ injection increased cellular ROS levels in WT hepatocytes but not in AQP3$^{-/-}$ hepatocytes. Oxidative stress, as detected by decreased glutathione (GSH)/glutathione disulfide (GSSG) ratio, was significantly increased in CCl$_4$-injected WT but not AQP3$^{-/-}$ liver homogenates (Fig. 1f). These findings support the involvement of AQP3 in the development of acute liver injury and oxidative stress in the CCl$_4$ model.

To investigate the role of AQP3 in macrophages and HSCs in liver injury, lethally irradiated WT and AQP3$^{-/-}$ mice (recipients) were reconstituted with bone marrow (BM) cells from WT and AQP3$^{-/-}$ mice (donors) in a CCl$_4$-induced acute liver injury model (Supplementary Fig. 1e). The chimeric mice produced by transferring WT BM cells into AQP3$^{-/-}$ recipients showed reduced CCl$_4$-induced increased expression of mRNA encoding TNF-α, CCL2, and α-SMA compared to the WT mice with WT BM transfer, suggesting that the development of liver injury requires AQP3 expression in non-hematopoietic cells such as HSCs and Kupffer cells. Transfer of AQP3$^{-/-}$ BM into WT or AQP3$^{-/-}$ recipients also reduced CCl$_4$-induced liver injury, suggesting the involvement of AQP3 expression in hematopoietic cells, most likely in recruited macrophages (Fig. 1g).

### AQP3-dependent macrophage activation is required for acute liver injury.
To investigate the involvement of macrophages in HSC activation and hepatic inflammation in CCl$_4$-induced acute liver injury, WT mice were administered clodronate liposomes, which depleted macrophages[33] as confirmed by anti-F4/80 immunostaining (Supplementary Fig. 2a). CCl$_4$-induced increases in TNF-α and α-SMA mRNA expression were suppressed by clodronate liposomes (Fig. 2a), as were hepatocyte ROS levels and oxidative stress (Fig. 2b, c), implicating the involvement of macrophages in HSC activation, increased hepatocyte ROS, and oxidative stress during acute liver injury.

We next investigated the requirement of AQP3 expression in macrophages for CCl$_4$-induced acute liver injury. Macrophages from WT and AQP3$^{-/-}$ bone marrow (Supplementary Fig. 2b) were injected intravenously 3 h before CCl$_4$ administration into AQP3$^{-/-}$ mice as described previously[34–36]. Adoptive transfer of

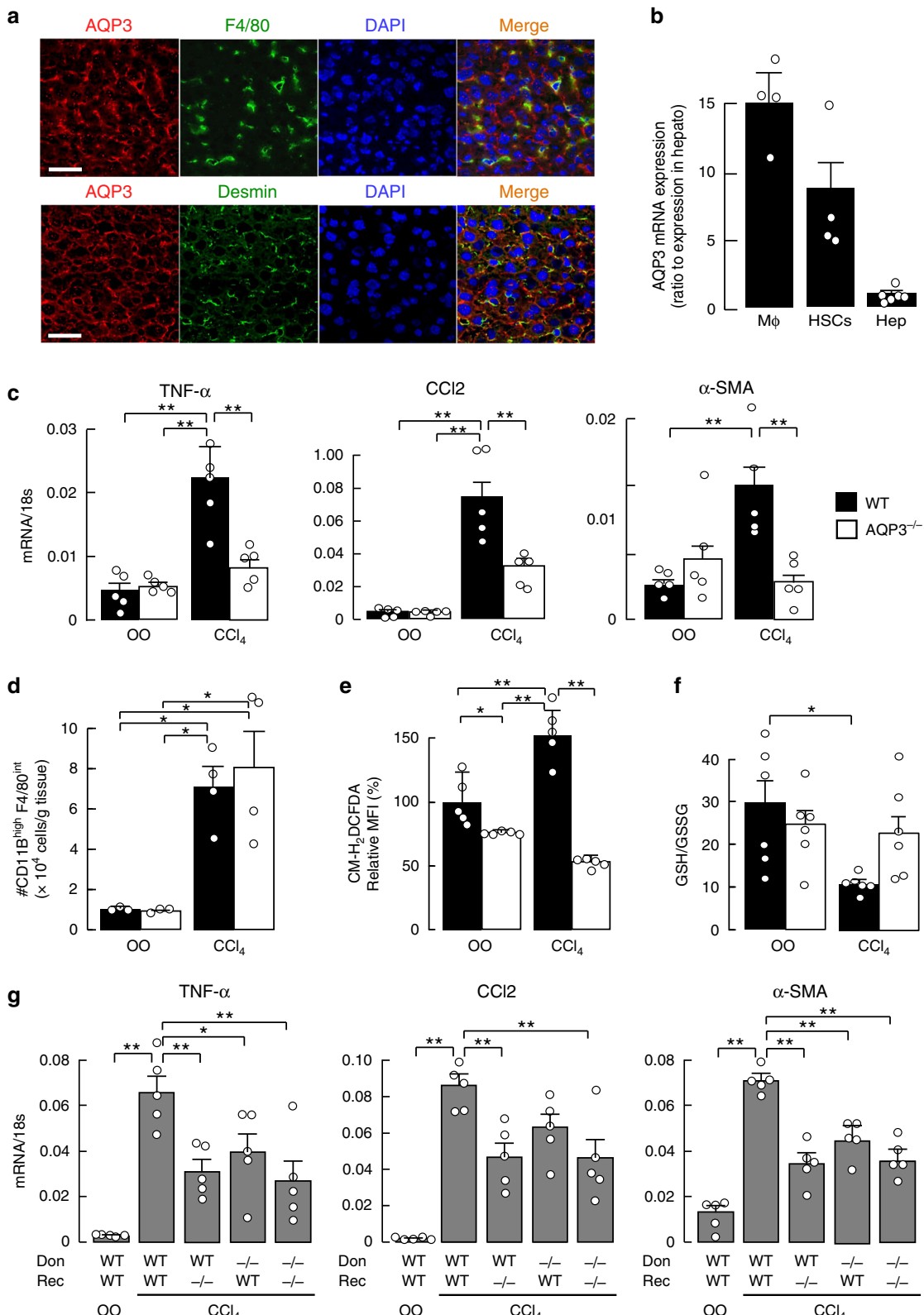

WT macrophages rescued the reduction in mRNA expression of TNF-α and α-SMA when compared with transfer of AQP3$^{-/-}$ macrophages (Fig. 2d). In addition, CCl$_4$-induced oxidative stress was greater with WT than AQP3$^{-/-}$ macrophage transfer (Fig. 2e), supporting the requirement of AQP3 in macrophages for CCl$_4$-induced liver injury.

Previous studies have shown that CCl$_4$-induced TNF-α production by activated macrophages contributes to the development of liver inflammation and fibrogenesis, which is dependent on NF-κB signaling and ROS status[1,5,28,29]. FACS analysis showed significantly greater TNF-α expression in CD11B$^+$ macrophages in WT than in AQP3$^{-/-}$ liver after CCl$_4$ injection, suggesting the

**Fig. 1 Reduced CCl4-induced acute liver injury in AQP3$^{-/-}$ mice. a** AQP3 immunofluorescence in mouse liver. Immunostaining with anti-AQP3 (Cy3, red), and anti-F4/80 (FITC, green, upper) or anti-desmin (FITC, green, lower). Bar, 100 μm. **b** AQP3 mRNA expression in hepatic macrophages (Mϕ), hepatic stellate cells (HSCs), and hepatocytes (Hep) from WT liver determined by real-time RT-PCR (mean ± SE, $n = 6$ for hepatocytes, $n = 4$ for macrophages and HSCs biologically independent samples). Data are expressed as the ratio to 18 s RNA. **c–f** CCl4 (1 ml/kg) or vehicle olive oil (OO) was injected intraperitoneally, and livers were excised at 24 h. **c** Expression of mRNAs encoding TNF-α, CCl2, and α-SMA by real-time RT-PCR (mean ± SE, $n = 5$ mice/group, **$p < 0.01$). Data are expressed as the ratio to 18 s RNA. **d** Dispersed liver cells were stained with anti-F4/80 and anti-CD11B, and analyzed by FACS, as shown in Supplementary Fig. 1d. Cell number of CD11B$^{high}$ F4/80$^{intermediate}$ hepatic macrophages (mean ± SE, $n = 3$ for OO, $n = 4$ for CCl4 mice/group, *$p < 0.05$). **e** Mean fluorescence intensity (MFI) of CM-H$_2$DCFDA by FACS analysis in hepatocytes (mean ± SE, $n = 5$ biologically independent samples *$p < 0.05$, **$p < 0.01$). **f** Ratio of GSH to GSSG in liver homogenate (mean ± SE, $n = 6$ mice/group, *$p < 0.05$). Statistical analysis for (**c**)–(**f**) was performed by two-way ANOVA with Tukey's multiple comparisons test. **g** Mice (WT or AQP3$^{-/-}$, age 8–10 weeks) were gamma-irradiated (900 rad) and injected intravenously with bone marrow from WT or AQP3$^{-/-}$ mice. The study was performed at 60 days after bone marrow transfer (mean ± SE, $n = 5$ mice/group, *$p < 0.05$, **$p < 0.01$ by one-way ANOVA with Dunnett's multiple comparisons test). Source data, including exact $p$ values, are provided as a Source data file.

involvement of AQP3 in TNF-α production by hepatic macrophages (Fig. 2f, Supplementary Fig. 2c). TNF-α production (assessed by ELISA assay) by isolated CD11B$^+$ hepatic macrophages from CCl4-treated WT mice was reduced by the NF-κB inhibitors BAY11-7082 and DHMEQ[37], or the ROS scavenger N-acetyl-L-cysteine (NAC) (Fig. 2g). These findings show that CCl4-induced TNF-α production by macrophages depends in part on NF-κB and ROS levels.

**Reduced H$_2$O$_2$ transport and NF-κB activation in AQP3$^{-/-}$ macrophages.** There is evidence that H$_2$O$_2$ plays an important role in the regulation of NF-κB[38,39]. We previously showed that AQP3-facilitated transport of H$_2$O$_2$ is involved in NF-κB activation as a secondary messenger in keratinocytes[23]. To study the involvement of AQP3 in NF-κB activation of macrophages during liver injury, sorted liver macrophages from naive WT or AQP3$^{-/-}$ mice were incubated with LPS for 24 h. Quantitative RT-PCR showed increased TNF-α expression in WT macrophages with LPS, which is related to inflammation in response to NF-κB activation, with little effect in AQP3$^{-/-}$ cells (Fig. 2h). We next investigated whether AQP3 could transport extracellular H$_2$O$_2$ into naive macrophages, as was found in keratinocytes, epithelial cells, and cancer cells[22–25,40–42]. Following extracellular addition of H$_2$O$_2$, intracellular H$_2$O$_2$ was significantly greater in WT than AQP3$^{-/-}$ macrophages within seconds (Fig. 2i). The increase in cellular ROS levels with LPS stimulation was also reduced in AQP3$^{-/-}$ macrophages (Fig. 2i).

In addition to liver macrophages, we confirmed the involvement of AQP3 in H$_2$O$_2$ transport and NF-κB activation in bone marrow-derived macrophages. Both H$_2$O$_2$ and LPS stimulation increased intracellular H$_2$O$_2$ level greater in WT compared to AQP3$^{-/-}$ cells (Fig. 2j). Moreover, both H$_2$O$_2$ and LPS stimulation induced p65 phosphorylation as seen by immunoblotting (Fig. 2k, Supplementary Fig. 2d) and p65 translocation into the nucleus as seen by immunofluorescence (Supplementary Fig. 2e), as markers of NF-κB activation in WT naive cells, which were reduced in AQP3$^{-/-}$ cells. Pretreatment with NAC and diphenyleneiodonium (DPI), a general Nox inhibitor, or incubation with catalase, which depletes extracellular H$_2$O$_2$, greatly suppressed LPS-induced NF-κB activation, suggesting that LPS-induced NF-κB activation in macrophages is partially dependent on extracellular ROS levels (Supplementary Fig. 2f). These results suggest that AQP3-mediated H$_2$O$_2$ transport regulates NF-κB activation in macrophages during acute inflammation.

**Activated macrophages are involved in HSC activation and hepatocyte oxidative stress.** In response to environmental signals, macrophages are polarized into M1 or M2 types[4–6]. M1 macrophages with an inflammatory phenotype are induced by

LPS and IFN-γ, and release pro-inflammatory cytokines (such as TNF, IL-1β), iNOS, and ROS. The inflammatory response with sustained LPS and IFN-γ stimulation, as seen by the increases in TNF-α and iNOS mRNA, was reduced in AQP3 deficiency (Fig. 3a). In contrast, differentiation of macrophages into the M2 subtype by IL-4 stimulation, quantified by arginase 1 (ARG1) as one of the M2 markers, was comparable in WT and AQP3$^{-/-}$ macrophages, while AQP3 expression level was greater in IL-4 treated macrophages than control or LPS/IFN-γ treated macrophages (Supplementary Fig. 3b, c). Cellular ROS levels were significantly higher in LPS/IFN-γ-treated inflammatory macrophages than in naive WT macrophages (Fig. 3b). H$_2$O$_2$ secretion from macrophages into the culture medium was remarkably greater in LPS/IFN-γ-treated than in naive macrophages, with much reduced secretion in AQP3$^{-/-}$ macrophages (Fig. 3c). Treatment with the AQP3 inhibitor AgNO$_3$ reduced H$_2$O$_2$ release from WT macrophages, suggesting AQP3-mediated H$_2$O$_2$ release (Fig. 3d). These findings support the conclusion that activated macrophages secrete H$_2$O$_2$ through AQP3.

Previous studies have shown that in early liver injury activated macrophages result in HSC transdifferentiation into myofibroblasts via TNF-α, as seen by the increase in α-SMA and collagen I expression, involving secretion of transforming growth factor-β (TGF-β)[1,5]. TNF-α-induced HSC activation in naive cells, as seen by the increased α-SMA and TGF-β1 expression, were similar in WT and AQP3$^{-/-}$ (Supplementary Fig. 3d), suggesting that HSC activation is less affected by AQP3.

To investigate whether activated macrophages cause HSC activation, isolated naive WT HSCs were co-cultured with CD11B$^+$ macrophages from CCl4-injected WT mice. Co-cultured HSCs with activated macrophages showed increased mRNA levels of α-SMA and TGF-β when compared with the levels in the absence of macrophages (Fig. 3e, Supplementary Fig. 3e). Pretreatment of either macrophages or HSCs with a blocking monoclonal TNF-α antibody or the NF-κB inhibitor DHMEQ significantly reduced α-SMA and TGF-β expression. Further, addition of NAC to the culture medium reduced α-SMA and TGF-β expression (Fig. 3e, Supplementary Fig. 3e). These findings suggest the requirement of activated macrophages for HSC activation during CCl4-induced acute liver injury, which is dependent on TNF-α, NF-κB cell signaling, and ROS. To support the requirement of macrophage AQP3 for HSC activation, we compared HSC activation in the same co-culture system with hepatic macrophages from WT or AQP3$^{-/-}$ mice. WT macrophages, but not AQP3$^{-/-}$ macrophages, increased TGF-β and α-SMA expression, indicating that AQP3-expressing macrophages play a pivotal role in HSC activation (Fig. 3f, Supplementary Fig. 3f).

We next tested the hypothesis that activated macrophages affect hepatocytes via AQP3-mediated H$_2$O$_2$ secretion. In co-cultures of WT hepatocytes with macrophages, LPS and IFN-γ-induced inflammatory WT macrophages increased cellular ROS

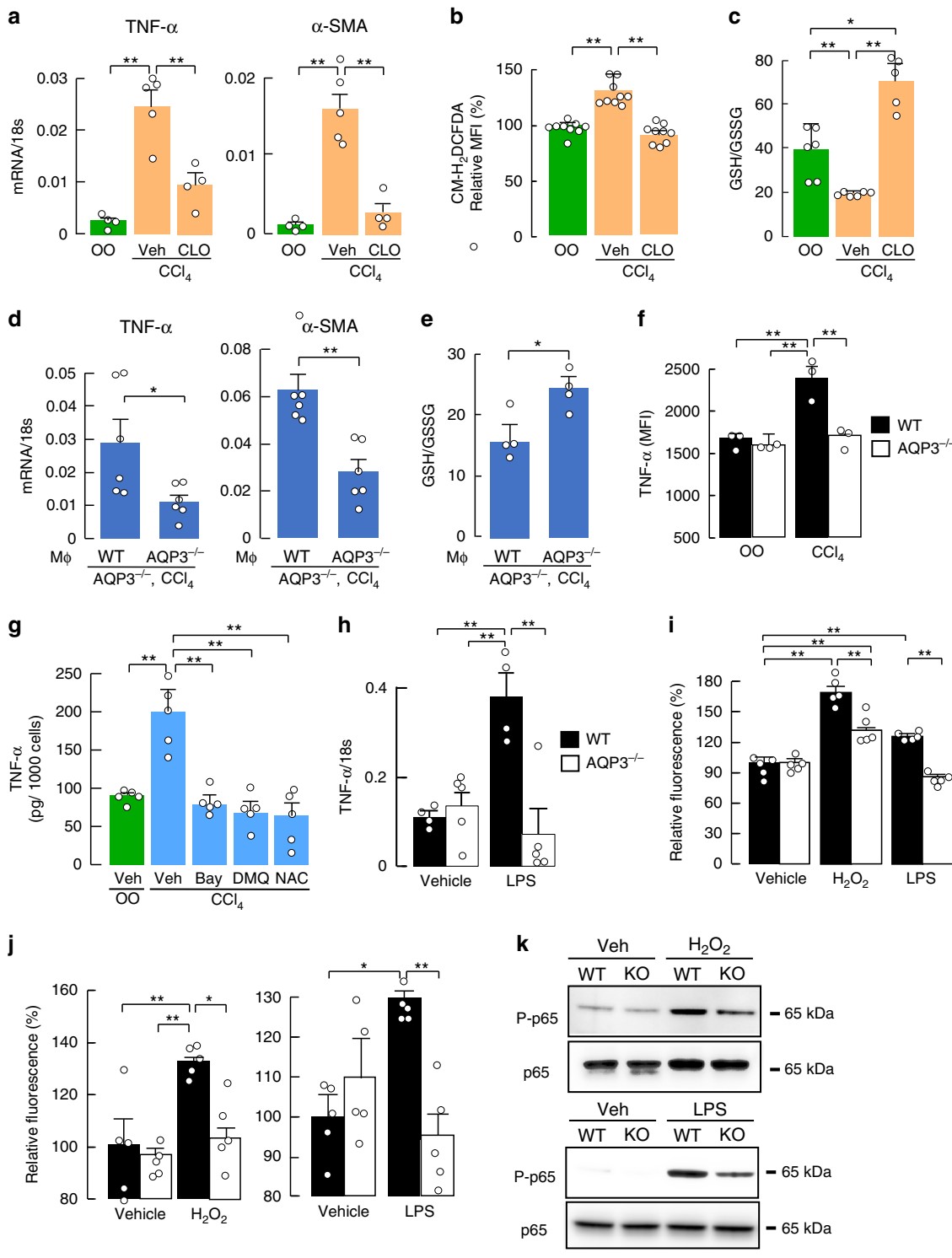

levels in hepatocytes when compared with naive control cells or AQP3$^{-/-}$ macrophages (Fig. 3g). Oxidative stress in hepatocytes was also increased by co-culture with activated WT but not with AQP3$^{-/-}$ macrophages, or NAC treatment (Fig. 3h). Taken together, these results support the involvement of macrophage AQP3 in HSC activation and hepatocyte oxidative stress during acute liver injury.

**Prevention of CCl$_4$-induced chronic liver injury in AQP3$^{-/-}$ mice.** To investigate the possible role of AQP3 in the pathogenesis

of chronic liver injury, CCl$_4$ was applied for 6 weeks to induce chronic liver injury and fibrosis in WT and AQP3$^{-/-}$ mice[43]. This model resulted in chronic liver injury with many necrotic cells and marked leukocyte infiltration on hematoxylin and eosin (H&E) staining (Fig. 4a, Supplementary Fig. 4a). The CCl$_4$ induced elevation in serum AST and ALT was lower in AQP3$^{-/-}$ mice (Supplementary Fig. 4b). Livers from WT mice showed advanced fibrosis by Sirius red staining (polymeric and monomeric collagen matrix) (Fig. 4b) and immunostaining with anti-α-SMA (Supplementary Fig. 4d), consistent with the previous reports[43].

**Fig. 2 AQP3-dependent macrophages activation is required for acute liver injury. a–c** Clodronate liposomes (CLO, 10 ml/kg) were injected intravenously, and injected $CCl_4$ intraperitoneally (1 ml/kg) or vehicle olive oil (OO) after 48 h. **a** mRNA expression of TNF-α and α-SMA in liver homogenates determined by real-time RT-PCR (mean ± SE, $n = 4$ for olive oil and clodronate/$CCl_4$, $n = 5$ for vehicle/$CCl_4$ mice/group, **$p < 0.01$). **b** MFI of CM-$H_2$DCFDA by FACS analysis in hepatocytes (mean ± SE, $n = 9$ biologically independent samples, **$p < 0.01$). **c** Ratio of GSH to GSSG in the liver homogenate (mean ± SE, $n = 6$ for olive oil and vehicle/$CCl_4$, $n = 5$ for clodronate/$CCl_4$ mice/group, *$p < 0.05$, **$p < 0.01$). Statistical analysis for (**a**)–(**c**) was performed by two-way ANOVA with Tukey's multiple comparisons test. **d**, **e** WT or AQP3$^{-/-}$ mouse-derived macrophages ($5 \times 10^6$ cells/200 μl PBS/head) were injected intravenously into AQP3$^{-/-}$ mice 1 h before $CCl_4$ injection. **d** mRNA expression of indicated genes in liver homogenates (mean ± SE, $n = 6$ mice/group, *$p < 0.05$, **$p < 0.01$ by two-tailed unpaired Student $t$-test). **e** Ratio of GSH to GSSG in liver homogenate (mean ± SE, $n = 4$ mice/group, *$p < 0.05$ by two-tailed unpaired Student $t$-test). **f** TNF-α expression in CD11B$^+$ F4/80$^+$ hepatic macrophage after $CCl_4$ injection or olive oil (OO) was analyzed by FACS as shown in Supplementary Fig. 2c. MFI of TNF-α (mean ± SE, $n = 3$ biologically independent samples, **$p < 0.01$ by two-way ANOVA with Tukey's multiple comparisons test). **g** TNF-α amount in culture medium by ELISA assay. CD11B$^+$ macrophages were magnetically isolated from WT liver with or without $CCl_4$ injection, and cultured with BAY11-7082 (20 μM), DHMEQ (1 μg/ml), or NAC (50 μM) (mean ± SE, $n = 5$ biologically independent samples, **$p < 0.01$ by one-way ANOVA with Dunnett's multiple comparisons test). **h**, **i** CD11B$^+$ macrophages were magnetically isolated from WT or AQP3$^{-/-}$ mouse liver. **h** Cells were treated with LPS (10 ng/ml) for 24 h. mRNA expression of TNF-α as the ratio to 18 s (mean ± SE, $n = 4$ for WT, $n = 5$ for AQP3$^{-/-}$ biologically independent samples, **$p < 0.01$). **i** $H_2O_2$ uptake into CD11B$^+$ liver macrophages. Cells were stimulated with $H_2O_2$ (30 μM) for 30 s or LPS (100 ng/ml) for 1 min, and cellular $H_2O_2$ was detected with CM-$H_2$DCFDA fluorescence using a plate reader (mean ± SE, $n = 5$ biologically independent samples, **$p < 0.01$). **j–k** Bone marrow-derived macrophages were generated from WT or AQP3$^{-/-}$ mice. **j** Cells were stimulated with $H_2O_2$ (30 μM) for 30 s or LPS (100 ng/ml) for 1 min. Cellular $H_2O_2$ was detected using CM-$H_2$DCFDA fluorescence using a plate reader (mean ± SE, $n = 5$ biologically independent samples, *$p < 0.05$, **$p < 0.01$). Statistical analysis for (**h**)–(**j**) was performed by two-way ANOVA with Tukey's multiple comparisons test. **k** Cells were incubated with $H_2O_2$ (300 μM) or LPS (100 ng/ml) for 30 min. Representative immunoblot using antibodies against phospho-p65 or p65. Source data, including exact $p$ values and uncropped immunoblot image, are provided as a Source data file.

Pathology was notably reduced in AQP3$^{-/-}$ mice, with less periportal necrosis and fibrosis (Fig. 4a, b, Supplementary Fig. 4a–c). AQP3 was found to be expressed in F4/80$^+$ macrophages in vehicle- and $CCl_4$-treated liver (Supplementary Fig. 4c).

RT-PCR showed that the $CCl_4$ injection increased fibrosis markers, including α-SMA, collagen type Iα, tissue inhibitor of metalloproteinase 1 (TIMP-1), and matrix metalloproteinase 9 (MMP9), in WT liver, with significantly lesser increases in AQP3$^{-/-}$ liver (Fig. 4c). Pro-inflammatory and pro-fibrogenic factors (TNF-α and TGF-β) were increased in WT liver, as previously reported[29], with reduced elevation in AQP3$^{-/-}$ liver (Fig. 4d). CD44 and CD24 expression, as cancer stem cell markers, were elevated in only WT liver with $CCl_4$ administration (Fig. 4e).

We next investigated the involvement of AQP3 in oxidative stress in the chronic liver injury model. Six weeks of $CCl_4$ treatment increased hepatic oxidative stress in WT liver, whereas AQP3$^{-/-}$ liver was unaffected (Fig. 4f). Also, the increase in DNA double-strand breaks, assessed from the number of phospho-H2AX-positive cells[44], was much reduced in AQP3$^{-/-}$ livers from $CCl_4$-treated mice (Fig. 4g), supporting the involvement of AQP3 in $CCl_4$-induced DNA damage. These findings indicate the requirement of AQP3 for the development of chronic liver injury with inflammation, high oxidative stress, and fibrosis.

**Monoclonal anti-AQP3 antibody inhibits $H_2O_2$ transport and macrophage activation**. The above findings suggest that inhibition of AQP3-facilitated $H_2O_2$ transport in macrophages might be of benefit in liver injury, though a suitable selective inhibitor has not been available[21]. To develop an anti-mouse monoclonal AQP3 antibody (anti-AQP3 mAb) that specifically recognizes the extracellular domain of AQP3, we generated a fragment (oligopeptide) composed of the amino acid sequence that corresponds to positions 148–157 in loop C of the extracellular domain, which are common to human and mouse AQP3. C57BL6 mice were immunized with this synthetic peptide together with mouse AQP3-overexpressing CHO-K1 cells. After selecting AQP3-binding colonies using several screening approaches, 10 IgG clones were identified (Supplementary Table 1).

Among the 10 IgG clones, four clones (mAb-C, E, H, and J) were found to bind specifically to 148–157 oligopeptides of loop C (Supplementary Fig. 5a). These clones showed different binding sites on loop C as analyzed by peptide ELISA (Supplementary

Tables 2 and 3). In CHO-K1 cells overexpressing human AQP3 (Supplementary Fig. 5b), significant binding was found of the four clones (Fig. 5a, Supplementary Fig. 5c). The anti-AQP3 mAbs significantly inhibited $H_2O_2$ uptake (Fig. 5b, Supplementary Fig. 5d, e) and glycerol uptake (Fig. 5c, Supplementary Fig. 5f, g) into AQP3-expressing cells. Some anti-AQP3 mAbs also weakly inhibited osmotically induced water transport in the AQP3-expressing CHO cells (Supplementary Fig. 5h). The apparently greater inhibitory effect on $H_2O_2$ and glycerol transport vs. water transport may be related to steric factors in the AQP3 pore. We also verified specific mAb binding to endogenous AQP3 in human keratinocytes (HaCaT), in which the binding was reduced by siRNA-mediated AQP3 knockdown (Supplementary Fig. 6a–c). The four anti-AQP3 mAbs inhibited $H_2O_2$ uptake into the cells to the same extent as with AQP3 knockdown (Supplementary Fig. 6d).

We next determined the effect of anti-AQP3 mAb on macrophage function. The binding of anti-AQP3 mAb-J to WT macrophages was greater than its binding to AQP3$^{-/-}$ macrophages (Fig. 5d). We also found binding, albeit relatively low, of mAb-J to HSC and hepatocytes (Supplementary Fig. 5i).

Cellular $H_2O_2$ uptake was reduced in WT macrophages incubated with anti-AQP3 mAbs (Fig. 5e). In addition, $H_2O_2$ secretion from LPS/IFN-γ-treated inflammatory macrophages was significantly reduced by anti-AQP3 mAb treatment (Fig. 5f). Importantly, treatment with anti-AQP3 mAb suppressed LPS-induced NF-κB activation (Fig. 5g, Supplementary Fig. 5j). In agreement with this observation, LPS/IFN-γ-induced increases in TNF-α and iNOS expression were reduced by anti-AQP3 mAb treatment (Fig. 5h). Treatment with anti-AQP3 antibody thus inhibits $H_2O_2$ transport and macrophage activation.

**Anti-AQP3 antibody prevented acute and chronic liver injury in mice**. We examined the effect of anti-AQP3 mAb on acute liver injury induced by $CCl_4$ (model in Fig. 1). Anti-AQP3 mAb or mouse monoclonal antibody (as control IgG) was administered intravenously (5 mg/kg weight) 1 day before $CCl_4$ injection. Anti-AQP3 remarkably suppressed the increases in serum AST and ALT (Fig. 6a). We verified the presence of administered fluorescein-conjugated anti-AQP3 mAb in the liver at 6–24 h after injection (Supplementary Fig. 7a). In addition, the increases in mRNAs encoding TNF-α, CCL2, α-SMA, and IL-6 in liver

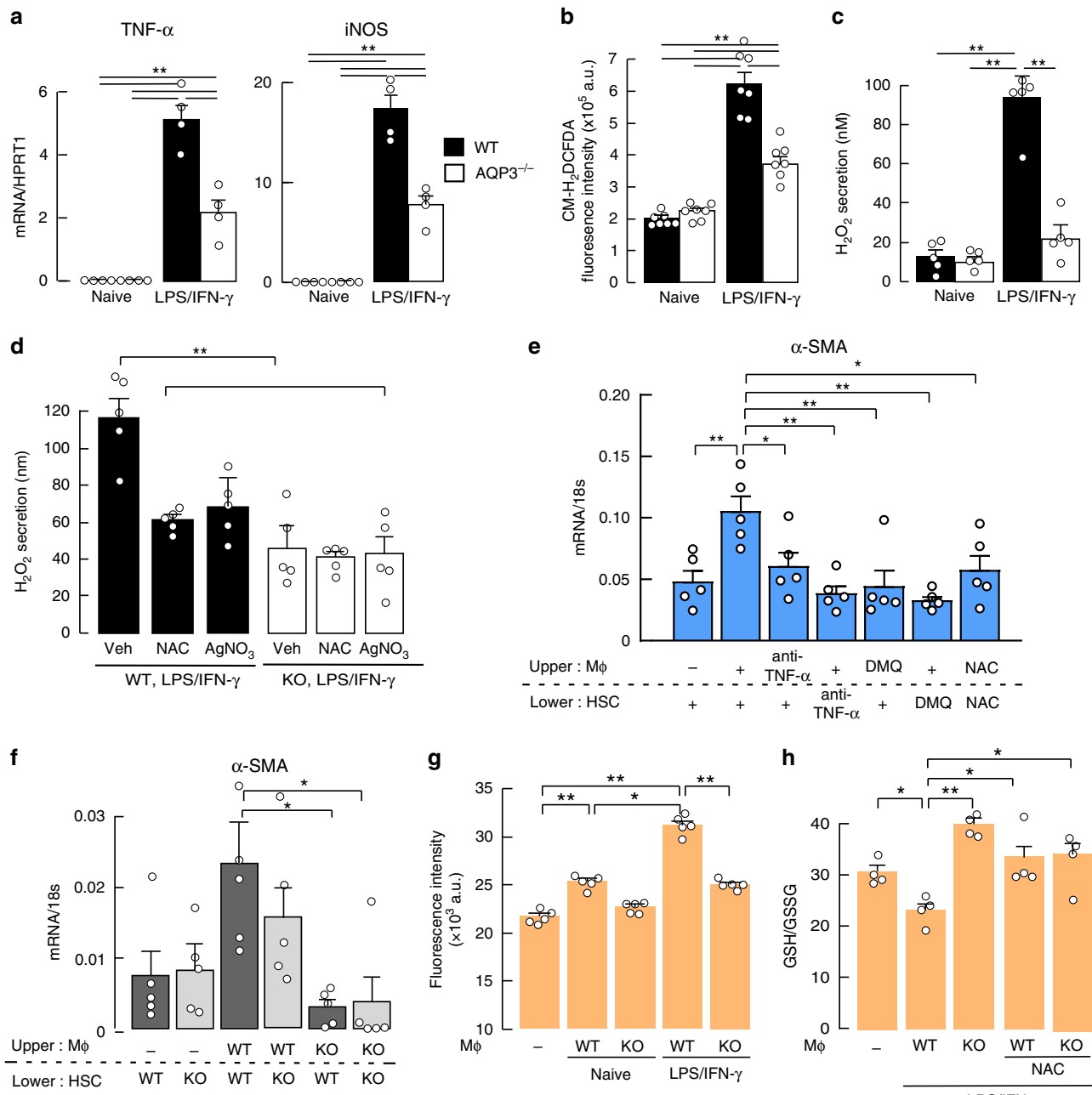

**Fig. 3 Activated macrophages are involved in HSC activation and hepatocyte oxidative stress. a–d** Naive bone marrow-derived macrophages from WT or AQP3$^{-/-}$ mice were treated with LPS/IFN-γ (1 ng/ml; 10 ng/ml) for 24 h. **a** mRNA expression of TNF-α and iNOS. Data are expressed as the ratio to HPRT1 (mean ± SE, $n = 4$ biologically independent samples, **$p < 0.01$). **b** Cellular ROS level (mean ± SE, $n = 7$ biologically independent samples, **$p < 0.01$). **c** $H_2O_2$ release into the culture medium (mean ± SE, $n = 5$ biologically independent samples, **$p < 0.01$). **d** $H_2O_2$ release to the culture medium from WT and AQP3$^{-/-}$ cells. After 24 h of LPS/IFN-γ treatment, cells were incubated with NAC (50 μM) or AgNO$_3$ (AQP3 inhibitor, 20 μM, 1 h), and $H_2O_2$ release to the culture medium measured (mean ± SE, $n = 5$ biologically independent samples, **$p < 0.01$). Statistical analysis for (**a**)–(**d**) was performed by two-way ANOVA with Tukey's multiple comparisons test. **e, f** HSCs from naive mice were co-cultured with hepatic macrophages from CCl$_4$-injected mice using a polycarbonate transwell membrane filter. **e** WT-derived HSCs or macrophages were incubated with anti-TNF-α (1 μg/ml) or DHMEQ (DMQ, 1 μg/ml) and then co-cultured. NAC (50 μM) was added to the culture medium. mRNA expression of α-SMA in HSCs (mean ± SE, $n = 5$ biologically independent samples, *$p < 0.05$, **$p < 0.01$ by one-way ANOVA with Dunnett's multiple comparisons test). **f** HSCs from WT or AQP3$^{-/-}$ mice were co-cultured with WT or AQP3$^{-/-}$ mouse-derived macrophages for 24 h. mRNA expression of α-SMA in HSCs (mean ± SE, $n = 5$ biologically independent samples, *$p < 0.05$ by one-way ANOVA with Dunnett's multiple comparisons test). **g, h** WT hepatocytes were co-cultured with macrophages from WT or AQP3$^{-/-}$ mice. Some macrophages were treated with LPS/IFN-γ (24 h) and subsequent co-culture (24 h). **g** Intracellular $H_2O_2$ in hepatocytes was monitored by CM-H$_2$DCFDA fluorescence (mean ± SE, $n = 5$ biologically independent samples, *$p < 0.05$, **$p < 0.01$). **h** Ratio of GSH to GSSG in hepatocytes (mean ± SE, $n = 4$ biologically independent samples, *$p < 0.05$, **$p < 0.01$). Statistical analysis for (**g**)–(**h**) was performed by one-way ANOVA with Tukey's multiple comparisons test. Source data, including exact $p$ values, are provided as a Source data file.

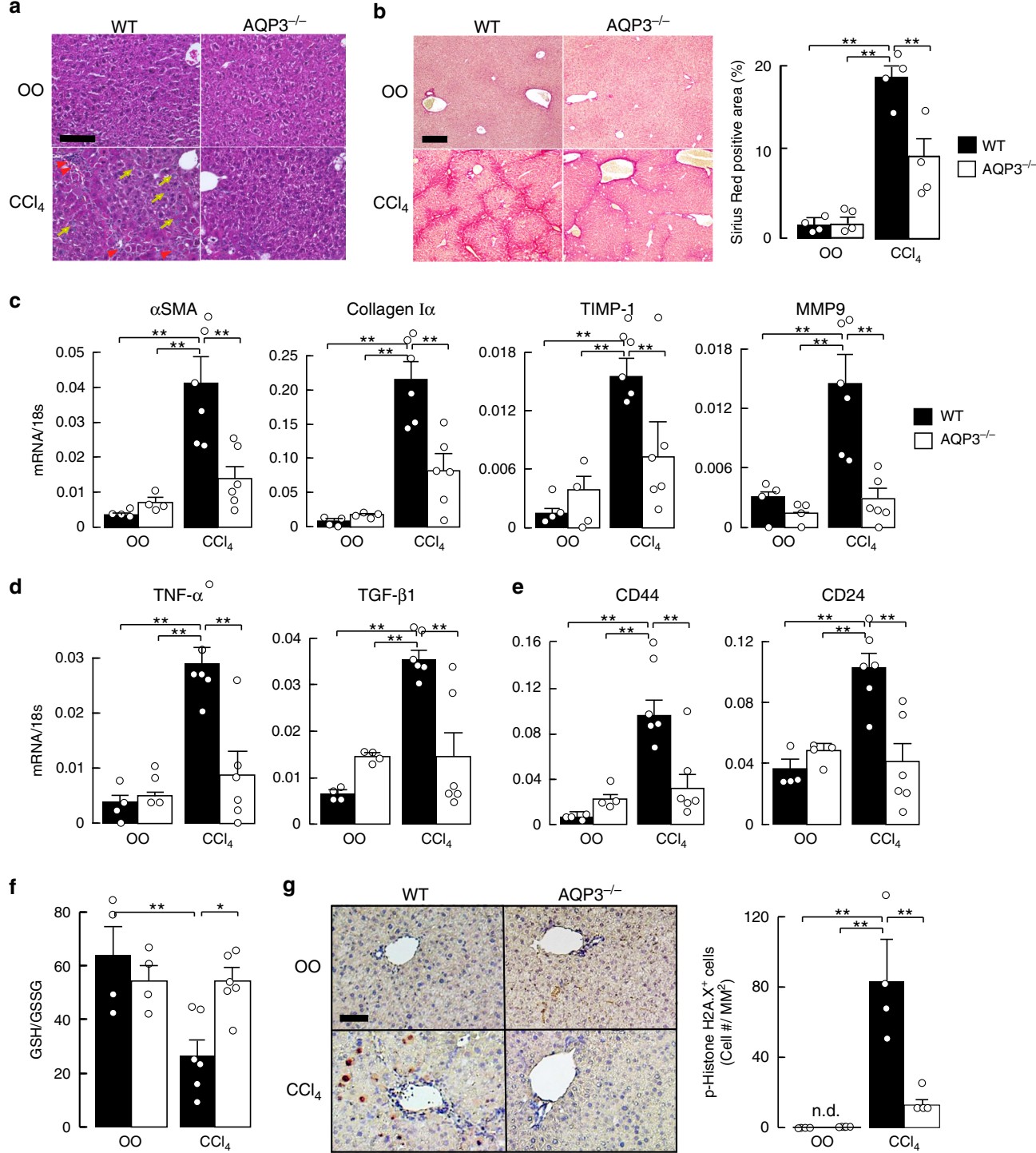

**Fig. 4 Reduced CCl4-induced chronic liver injury in AQP3−/− mice. a–g** CCl4 (0.5 ml/kg) or vehicle olive oil (OO) was intraperitoneally injected twice a week for 6 weeks. **a** Hematoxylin and eosin staining of liver from WT and AQP3−/− mice. Bar, 100 μm. Arrow: infiltrating lymphocytes. **b** (left) Sirius red staining. Bar, 200 μm. (right) Sirius red-positive staining area (mean ± SE, n = 4 sections from 4 individual mice, **p < 0.01). **c–e** mRNA expression of indicated genes in liver homogenates by real-time RT-PCR (mean ± SE, n = 4 for olive oil, n = 6 for CCl4 mice/group, **p < 0.01). Data are expressed as the ratio to 18s RNA. **f** Ratio of GSH to GSSG in liver homogenate (mean ± SE, n = 4 for olive oil, n = 6 for CCl4 mice/group *p < 0.05, **p < 0.01). **g** (left) Phospho-histone H2A.X staining. Bar, 200 μm. (right) Number of phospho-histone H2A.X-positive stained cells (mean ± SE, n = 4 sections from four individual mice, >100 cells from over two different fields from one mouse, **p < 0.01). Statistical analysis for (**b**)–(**g**) was performed by two-way ANOVA with Tukey's multiple comparisons test. Source data, including exact p values, are provided as a Source data file.

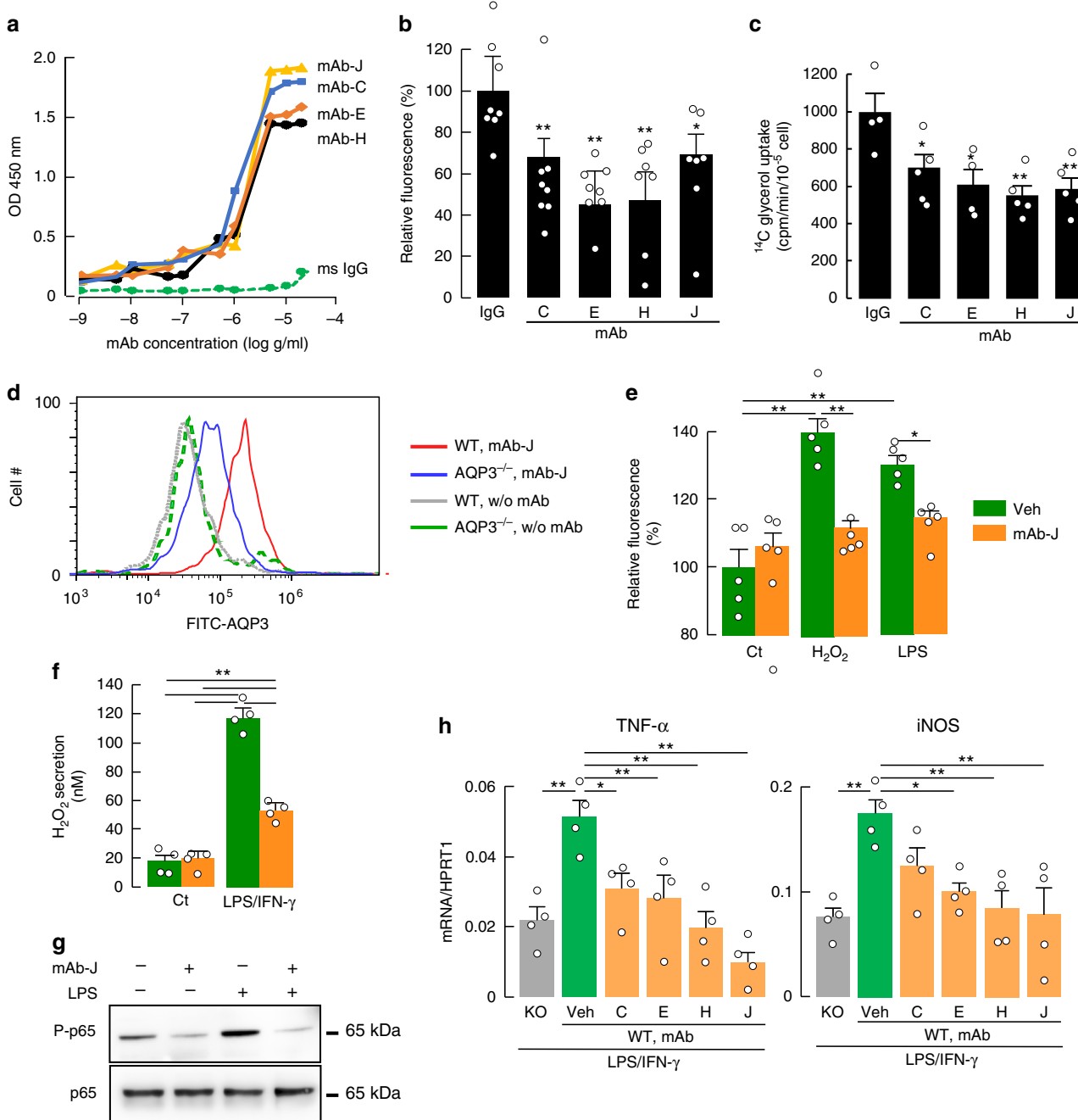

**Fig. 5 Anti-AQP3 mAb inhibits $H_2O_2$/glycerol transport and macrophage activation. a** Binding of anti-AQP3 mAb (C, E, H, and J) to human AQP3-expressing CHO-K1 cells. Experiments were performed in duplicate and repeated three times with similar results. **b**, **c** Effect of mAb on $H_2O_2$ uptake (**b**) or glycerol uptake (**c**) in human AQP3-expressing CHO-K1 cells. Cells were incubated with mAb (1 µg/ml, 37 °C) or control monoclonal anti-mouse IgG prior to transport measurements. **b** Cellular $H_2O_2$ measured after adding $H_2O_2$ (40 µM) using CM-H2DCFDA fluorescence (mean ± SE, $n = 8$ for IgG, C and E, $n = 7$ for H and J biologically independent samples, *$p < 0.05$, **$p < 0.01$ vs. mouse IgG treated by one-way ANOVA with Dunnett's multiple comparisons test). **c** Glycerol uptake. Cellular [$^{14}$C] radioactivity was measured after adding [$^{14}$C]-glycerol for 1 min (mean ± SE, $n = 4$ for IgG and E, $n = 5$ for C, H, and J biologically independent samples, *$p < 0.05$, **$p < 0.01$ vs. mouse IgG treated by one-way ANOVA with Dunnett's multiple comparisons test). **d**–**h** Bone marrow-derived macrophages generated from WT or AQP3$^{-/-}$ mice were used. **d** Flow cytometric analysis of binding of anti-AQP3 mAb-J in WT or AQP3$^{-/-}$ macrophages. **e** Cellular $H_2O_2$ after addition of $H_2O_2$ (30 µM, 30 s) or LPS (100 ng/ml, 1 min) in WT macrophages treated with anti-AQP3 mAb-J (100 ng/ml, 1 h, mean ± SE, $n = 5$ biologically independent samples, *$p < 0.05$, **$p < 0.01$ by two-way ANOVA with Tukey's multiple comparisons test). **f** $H_2O_2$ release to the culture medium from WT macrophages treated with anti-AQP3 mAb-J (100 ng/ml, 1 h). WT Macrophages were incubated with LPS/IFN-γ (1 ng/ml; 10 ng/ml) for 24 h (mean ± SE, $n = 4$ biologically independent samples, **$p < 0.01$ by two-way ANOVA with Tukey's multiple comparisons test). **g** Representative immunoblot using antibodies against phospho-p65 and p65. WT macrophages were incubated with anti-AQP3 (mAb-J, 1 µg/ml, 1 h) and thereafter stimulated with LPS (100 ng/ml, 30 min). **h** WT or AQP3$^{-/-}$ (KO) macrophages were treated with LPS/IFN-γ (1 ng/ml; 10 ng/ml) for 24 h. Some WT cells were co-incubated with anti-AQP3 mAb (C, E, H, and J, 1 µg/ml). mRNA expression of TNF-α and iNOS. Data are expressed as the ratio to HPRT1 (mean ± SE, $n = 4$ biologically independent samples, *$p < 0.05$, **$p < 0.01$ vs. WT/vehicle by one-way ANOVA with Dunnett's multiple comparisons test). Source data, including exact $p$ values and uncropped immunoblot image, are provided as a Source data file.

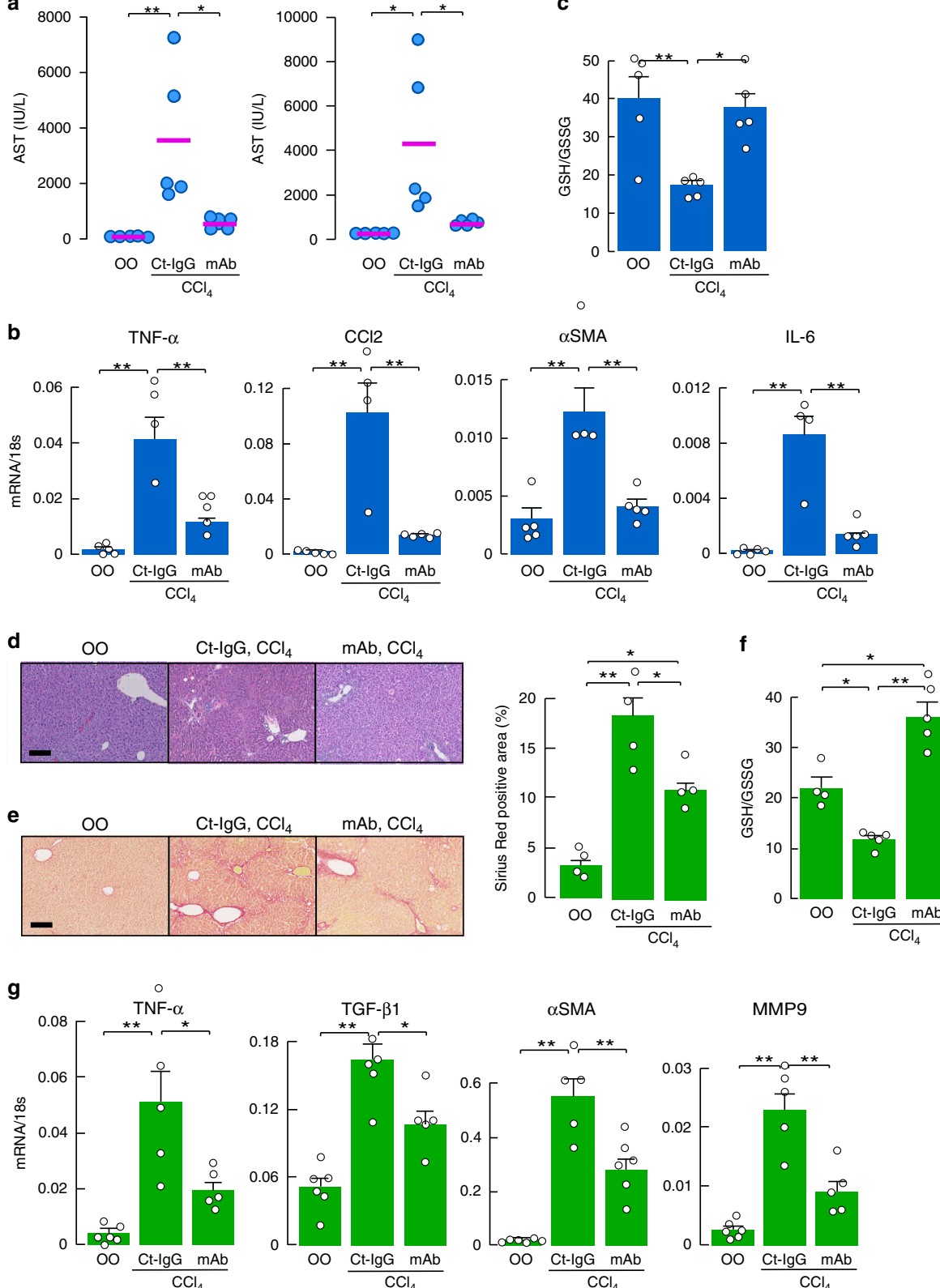

homogenates were greatly suppressed by anti-AQP3 mAb (Fig. 6b), as was CCl₄-induced oxidative stress (Fig. 6c). Liver injury was not affected by administration of the anti-AQP3 mAb in AQP3$^{-/-}$ mice with CCl₄ injection (Supplementary Fig. 7b, c).

We also investigated the effect of anti-AQP3 mAb on CCl₄-induced chronic liver injury (model in Fig. 4). Administration of

anti-AQP3 mAb prevented chronic liver injury, with fewer necrotic cells observed on H&E staining (Fig. 6d) and much reduced fibrosis with Sirius red staining (Fig. 6e). Anti-AQP3 mAb suppressed CCl₄-induced hepatic oxidative stress (Fig. 6f) and the increases in mRNA encoding inflammatory cytokines and fibrosis-related markers (Fig. 6g).

**Fig. 6 Prevention of acute and chronic liver injury by anti-AQP3 mAb. a–c** Effect of anti-AQP3 mAb on acute liver injury. Anti-AQP3 (mAb-J, 10 mg/kg weight, PBS) or control monoclonal anti-mouse IgG (Ct-IgG, 10 mg/kg weight, PBS) was injected intravenously 1 day before $CCl_4$ injection (1 ml/kg weight, olive oil; OO). Liver and blood were collected at 24 h. **a** Serum AST and ALT (mean ± SE, $n = 5$ mice/group, $*p < 0.05$, $**p < 0.01$). **b** mRNA expression of indicated genes by real-time RT-PCR (mean ± SE, $n = 4$ for control IgG/$CCl_4$, $n = 5$ for olive oil and anti-AQP3 mAb/$CCl_4$ mice/group, $**p < 0.01$). Data are expressed as the ratio to 18s RNA. **c** Ratio of GSH to GSSG in the liver homogenate (mean ± SE, $n = 5$ mice/group, $*p < 0.05$, $**p < 0.01$). **d–g** Effect of anti-AQP3 on chronic liver injury. $CCl_4$ (0.5 ml/kg) or vehicle olive oil (OO) was intraperitoneally injected twice a week for 4 weeks. Anti-AQP3 mAb (mAb-J, 10 mg/kg, PBS) or control monoclonal anti-mouse IgG (Ct-IgG, 10 mg/kg, PBS) was intravenously injected 1 day before each $CCl_4$ injection. **d** Hematoxylin and eosin staining of liver. Bar, 100 μm. **e** Sirius red staining. Bar, 200 μm. (right) Sirius red-positive staining area (mean ± SE, $n = 4$ sections from four individual mice, $*p < 0.05$, $**p < 0.01$). **f** Ratio of GSH to GSSG in the liver homogenate (mean ± SE, $n = 4$ for olive oil, $n = 5$ for $CCl_4$ mice/group, $*p < 0.05$, $**p < 0.01$). **g** mRNA expression of indicated genes in liver homogenates by real-time RT-PCR. Data are expressed as the ratio to 18s (mean ± SE, $n = 6$ for olive oil, $n = 5$ for $CCl_4$ mice/group, $*p < 0.05$, $**p < 0.01$). Statistical analysis for (**a**)–(**g**) was performed by one-way ANOVA with Tukey's multiple comparisons test. Source data, including exact $p$ values, are provided as a Source data file.

The effect of anti-AQP3 mAb was also studied in two additional experimental mouse models of liver injury. One model involves administration of thioacetamide (TAA), which is metabolized to TAA sulfine, producing liver cell damage, inflammation, and subsequently liver fibrosis[45–50]. Histological examination showed more necrotic hepatocytes in WT compared to AQP3$^{-/-}$ liver following intraperitoneal injection of TAA (Supplementary Fig. 8a). TAA injection increased mRNA expression of inflammatory factors TNF-α and CCL2 in WT liver (at 24 and 72 h), as well as fibrogenic markers α-SMA and collagen type Iα (72 h), each of which was significantly reduced in AQP3$^{-/-}$ liver (Supplementary Fig. 8b). Administration of anti-AQP3 mAb one day before TAA injection suppressed necrosis as shown by H&E staining (Supplementary Fig. 8c), and reduced TNF-α and CCL2 mRNA expression at 24 h (Supplementary Fig. 8d). The anti-AQP3 mAb also suppressed TAA-induced hepatic oxidative stress (Supplementary Fig. 8e).

As a third model, we tested the azoxymethane (AOM) injection model, a well-established mouse model of colon carcinogenesis associated with liver injury[51–56]. We found that AOM-induced acute and chronic liver injury was reduced in AQP3$^{-/-}$ mice compared with WT mice (Supplementary Fig. 9a–c for chronic model; Supplementary Fig. 9d for acute model). Administration of the anti-AQP3 mAb before AOM injection reduced acute liver injury, as shown by reduced AST/ALT and oxidative stress, with reduced TNF-α, CCL2, and α-SMA (Supplementary Fig. 9e–g).

## Discussion

A neutralizing anti-AQP3 mAb was generated that bound to the extracellular domain of AQP3 in macrophages and inhibited AQP3-mediated $H_2O_2$ transport and NF-κB cell signaling. Administration of the neutralizing anti-AQP3 mAb to mice suppressed inflammation and liver injury in $CCl_4$-, TAA-, and AOM-induced models of liver injury. The anti-AQP3 mAb might block the AQP3 channel directly or neutralize its function by an allosteric mechanism. We note that a relatively high concentration of mAb was used, as the antibody was not optimized for pharmacological properties. Notwithstanding these considerations, our findings provide evidence for anti-AQP3 mAb as a novel approach for inhibition of hepatic macrophage function in liver injury. There is increasing evidence from AQP3$^{-/-}$ mice and AQP3-knockdown cells for the involvement of AQP3 in various inflammatory diseases including atopic dermatitis, psoriasis, allergy, and cancer progression, in which AQP3 transport function supported cell proliferation, migration, and inflammation[22–26,42,57,58]. Also, many descriptive studies have shown positive correlations between AQP3 expression and cancer progression and prognosis[59–65]. Thus, AQP3 inhibition has been suggested as a potential therapeutic target for a variety of diseases. However, as most AQP3 inhibitors contain the metals mercury, copper, or gold[21], there are not at present suitable drug-like small molecule AQP3 inhibitors.

We found evidence for the requirement of AQP3 for the development of liver injury and fibrosis, which involved AQP3-mediated $H_2O_2$ transport in macrophages. In a mouse model of acute liver injury induced by a single injection of $CCl_4$, increases in TNF-α production, α-SMA expression, and oxidative stress were greatly reduced in liver of AQP3$^{-/-}$ mice in a manner dependent on hepatic macrophages. The reduction in acute liver injury in AQP3$^{-/-}$ mice was rescued by the transfer of WT but not AQP3$^{-/-}$ macrophages, providing evidence for the requirement of AQP3 expression in macrophages for the early inflammatory response in the liver. In vitro studies showed that AQP3-mediated $H_2O_2$ uptake was involved in NF-κB cell signaling in naive macrophages, resulting in their activation. Moreover, excess $H_2O_2$ produced by inflammatory macrophages was transported extracellularly via AQP3, affecting surrounding cells including HSCs and hepatocytes. Consequently, in AQP3 deficiency there is reduced activation of naive macrophages and HSCs resulting in impaired inflammation and profibrotic responses. The reduced early inflammation and fibrogenesis may be responsible for the greatly reduced chronic liver injury and fibrosis in AQP3$^{-/-}$ mice. These findings support the involvement of AQP3 in the development of liver injury through a mechanism involving its $H_2O_2$ transport function and macrophage activation, and suggest AQP3 as a novel potential therapeutic target in liver injury.

An expanding body of literature supports an important biological effect of $H_2O_2$ on cellular functions[66,67]. The transient accumulation of intracellular $H_2O_2$ near the plasma membrane, though relatively small, modulates specific cell signaling pathways to control a diverse set of physiological functions, including cell proliferation, differentiation, and migration[14,68]. Higher concentrations of $H_2O_2$ produced by cancer cells or immune cells induces oxidative stress, leading to inflammation or cytotoxicity[69,70]. Oxidative stress has been linked to a myriad of pathologies including initiation and progression of liver injury[10,11,71]. In this study, we showed that inflammatory macrophages produced much more intracellular $H_2O_2$ compared to the quiescent cells, and hence could release larger quantities of $H_2O_2$ into extracellular space through AQP3. The increased extracellular $H_2O_2$ affected surrounding liver cells, including HSCs and hepatocytes, amplifying the liver injury (Supplementary Fig. 9). Our data support a liver injury mechanism in which AQP3 mediates crosstalk between macrophages and neighboring cells via its $H_2O_2$ transport function. Prevention of this cell–cell pro-inflammatory communication by AQP3 inhibition represents a potential novel therapeutic approach for oxidative stress-related diseases.

In summary, our findings support the novel role for AQP3 in the pathogenesis of liver injury and fibrosis in which AQP3-mediated intracellular $H_2O_2$ uptake is required for NF-κB cell signaling and subsequent macrophage activation. These findings thus support AQP3 inhibition as a potential therapeutic approach

for liver injury and offer a monoclonal antibody approach to accomplish this.

## Methods

**Mice.** C57BL/6 mice were purchased from Japan SLC, Inc. AQP3$^{-/-}$ mice (C57BL/6 genetic background) were generated by targeted gene disruption[72]. Seven- to 10-week-old mice were used. All animal experiments were approved by the President of Keio University, following the consideration by the Institutional Animal Care and Use Committee of Keio University (approval no: 16075) and by Genetic Modification Safety Committee, Keio University School of Medicine (approval no. 28-029), and were carried out in accordance with institutional procedures, national guidelines, and the relevant national laws on the protection of animals.

**Development of an anti-AQP3 antibody.** An oligopeptide was synthesized consisting of the amino acid sequence corresponding to positions 148–157 of the mouse/human AQP3 polypeptide. A C57BL/6 mouse was immunized by the synthetic peptide together with mouse AQP3-overexpressing CHO-K1 cells and an adjuvant. Four weeks later, immune cells were collected from the immunized mouse and an antibody gene phage library was constructed[73]. After several screenings, selected AQP3-binding colonies were made into IgG immunoglobulins to give ten anti-AQP3 mAbs.

**Mouse models of acute and chronic liver injury by CCl4.** To induce liver injury and fibrosis, carbon tetrachloride (CCl$_4$, 0.5 ml/kg weight) or vehicle control (olive oil) was intraperitoneally injected twice weekly for 6 weeks. For acute injury, mice received an intraperitoneal injection of CCl$_4$ (1 ml/kg weight). Mice were sacrificed at 24 h after the final CCl$_4$ injection. To deplete macrophages in some experiments, mice were intravenously administered chlodronate liposomes (10 μl/g weight, Xygieia Bioscience) 2 days before CCl$_4$ injection. In some studies, anti-AQP3 mAb or mouse monoclonal antibody (as a control IgG) was administered intravenously (5 mg/kg weight) 1 day before each CCl$_4$ injection.

**Bone marrow transplantation.** For bone marrow (BM) transplantation, red blood cells from WT and AQP3$^{-/-}$ BM cells were subjected to hypotonic cell lysis. WT and AQP3$^{-/-}$ recipients (8–10 weeks old) were γ-irradiated with doses of 900 rad. After irradiation, the mice received 10$^6$ BM cells intravenously. This protocol consistently gave >95% reconstitution of the recipient by donor hematopoietic cells, as evaluated by separate transplantation experiments using BM from C57BL/6-CD45.1 congenic mice (Supplementary Fig. 1e).

**Primary culture of mice hepatocytes, hepatic macrophage, and HSC.** Livers were perfused with liver perfusion medium (Thermo Fisher Scientific, #17701-038) and then liver digest medium (Thermo Fisher Scientific, #17703-034) according to manufacturer's instructions[74]. The dispersed cells were centrifuged (50 × $g$, 1 min) to recover nonparenchymal cells-containing supernatants and pelleted hepatocytes. Isolated hepatocytes were plated on collagen I-coated plate (BD Biosciences) and cultured with HepatoZYME-SFM (Thermo Fisher Scientific, #17705-021). To obtain HSCs and macrophages, recovered nonparenchymal cells were isolated by density gradient centrifugation with Optiprep solution (11.5 and 15%) (Axis Shield, Dundee, UK) diluted with HBSS[75,76]. In some experiments, recovered non-parenchymal cells were sorted using anti-F4/80 or anti-CD11B microbeads with MACS separator according to manufacturer's instructions (Milteny Biotec). Macrophages were co-cultured with HSCs or hepatocytes using a polycarbonate transwell membrane filter (0.4-μm pore size, Corning Costar, Cambridge, MA).

**Bone marrow-derived macrophage preparation.** Single-cell suspensions of bone marrow cells were collected from femur and tibia, and cultured in RPMI 1640 (Invitrogen) containing 10 ng/ml GM-CSF (PeproTech Inc.), 10% FBS, 50 μM 2-mercaptoethanol, 2 mM L-glutamine, 25 mM HEPES, 1 mM nonessential amino acids, 1 mM sodium pyruvate, 1% penicillin-streptomycin for at least 6 days. More than 90% of cultured cells were confirmed as macrophages by FACS analysis (Supplementary Fig. 2b). To activate macrophages, cells were incubated with LPS (1 ng/ml) and IFN-γ (10 ng/ml) for over 1 day.

**Cell culture and transfection.** CHO-K1 cells (Riken BRC cell bank, Japan) and human keratinocyte HaCaT cells (Cell Line Service, Germany) were grown in Dulbecco's Modified Eagle Medium containing 10% FBS and 1% penicillin-streptomycin. The cDNA plasmid for human AQP3 (pCMV6 vector, Origine) was transfected into CHO-K1 with TransIT-X2 (Mirus) and positive cells were obtained by G418 selection. HaCaT cells were transfected using Lipofectamine 2000 (Invitrogen) with AQP3- or non-targeting-siRNA (ON-TARGET plus SMART pool, Thermo Scientific). We generally found that 70–80% of cells become positive after transfection.

**RNA extraction and real-time quantitative RT-PCR.** Total RNA was extracted using TRIZOL (Invitrogen). The cDNA was reverse transcribed from total RNA using the Prime Script RT reagent kit (Takara Bio, Otsu, Japan). Quantitative RT-PCR was performed using SYBR Green I (Takara Bio) and StepOne Plus real-time PCR apparatus (Thermo Fisher Scientific).

**Immunohistochemistry and immunofluorescence.** Paraffin-embedded sections were stained with hematoxylin and eosin, Sirius red (ScyTek Laboratories Inc.), or anti-phospho-Histone H2A.X (Cell Signaling, 1:400) with biotinylated IgG and horseradish peroxidase-conjugated ABC reagent (Vector Laboratories, Burlingame, CA). The positive area of Sirius red staining and the number of brown color stained-H2A gamma positive cells were analyzed by Tissue Quest (TissueGnostics). Frozen sections were fixed with cold acetone, and immunostained with anti-AQP3 (Millipore, 1:100), anti-F4/80 (eBioScience, 1:400), or anti-desmin (R&D Systems, 10 μg/ml) antibodies.

**Assays of cellular H2O2, extracellular H2O2, and GSH/GSSG.** Cellular H$_2$O$_2$ was assayed using the CM-H2DCFDA reagent (Invitrogen) according to manufacturer's instructions using flow cytometry (Gallios, Beckman, CA, USA) on a microplate reader (SpectraMax i3x; Molecular Devices). To determine cell-derived ROS level, accumulated ROS in the medium was quantified by OxiSelect$^{TM}$ Hydrogen Peroxide Assay Kit (Cell Biolabs, Inc.). For GSH/GSSG analysis, liver or cells were homogenized with 5% 5-sulfosalicylic acid dihydrate, and GSSG and GSH levels were measured using a GSSG/GSH Quantification kit (Dojindo, Japan) according to the manufacturer's instructions.

**Glycerol uptake assay.** Cells were incubated with [$^{14}$C]-labeled glycerol (Perkin Elmer, NEC441x, 2 μCi/ml PBS containing 10 mM glycerol) for 1 min at room temperature. After washing cells three times with ice-cold PBS, cells were disrupted with 1 M NaOH. Cell-associated radioactivity was determined by scintillation counting.

**Immunoblotting.** Cells were lysed with RIPA buffer (Cell Signaling Technology) and the supernatant (10,000 × $g$, 10 min, 4 °C) was used for immunoblotting with antibodies against phospho-p65 and p65 (Cell Signaling Technology, 1:1000). A horseradish peroxidase-conjugated secondary anti-rabbit antibody (Cell Signaling Technology, 1:1000) was used and visualized by chemiluminescence (GE Healthcare).

**AQP3-binding assay.** Human AQP3-expressing CHO-K1 or HaCaT cells plated on 96-well plates were incubated with AQP3 mAbs (0.1 ng – 1 μg/ml, 1 h, 4 °C). After washing cells, bound antibody was detected with HRP-conjugated anti-mouse IgG (Cell Signaling Technology, 1:1000) and TMB (eBioScience), followed by stop reagent. Absorbance (450 nm) was measured on a plate reader (SpectraMax i3x, Molecular Devices).

**Statistical analysis.** Statistical analysis was performed using the two-tailed Student's $t$-test, one-way, or two-way ANOVA by GraphPad Prism8.

**Reporting summary.** Further information on research design is available in the Nature Research Reporting Summary linked to this article.

## Data availability

All other relevant data supporting the key findings of this study are available within the article and its Supplementary Information file or from the corresponding authors upon request. Source data are provided with this paper.

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

## Acknowledgements

We thank Drs. Hiroki Satooka, Sachiko Watanabe, and Catharina Sagita Moniaga for supporting preliminary experiments. We thank Drs. Shu Narumiya, Noriyuki Morikawa, and Yoshiaki Morita, and Center for Innovation in Immunoregulative Technology and Therapeutics, Graduate School of Medicine, Kyoto University for supporting development of anti-AQP3 antibody. We thank Drs. Maruyama and Okumura for supporting the experiment using AQP3 peptide mutants. We thank Dr. Hayato Takahashi for providing CD45.1 mice. We thank Dr. Kazuo Umezawa for providing DHMEQ reagent. This work was supported in part by grants from Astellas Pharma Inc. in the Creation of Innovation Centers for Advanced Interdisciplinary Research Areas Program (M.H.-C.), the Princess Takamatsu Cancer Research Fund (15-24717, M.H.-C.), Keio University Academic Development Funds (M.H.-C.), Suntory Global Innovation Center Ltd. program "Water Channeling Life" (M.Y.), and U.S. National Institutes of Health grant DK72517 and EY13574 (A.S.V.).

## Author contributions

M.H.-C. conceived the study and all authors were involved in the design of experiments. M.H.-C. and M.T. performed the experiments. M.H.-C., M.T., and M.Y. analyzed the data. M.H.-C. and A.S.V. wrote the manuscript.

## Competing interests

The authors declare no competing interests.
