## [Peer Review File · Nature Communications]

Reviewers' Comments:

Reviewer #1:

Remarks to the Author:

In this manuscript Hara-Chikuma et al. examine the role of aquaporin-3 (AQP3) and its transport of hydrogen peroxide in liver injury and fibrosis. Using AQP3 knockout mice with appropriate strain background controls, these authors show first that in wild-type mice AQP3 was expressed in macrophages and hepatic stellate cells in the liver (with low expression in hepatocytes) and that carbon tetrachloride induced a lesser increase in pro-inflammatory cytokines (TNF α and Ccl2), alpha-SMA (a marker of myofibroblasts) and oxidative stress, as well as a reduction in the ratio of reduced to oxidized glutathione (GSH/GSSG), despite having similar infiltration of macrophages into the liver. In wild-type mice they then showed the importance of macrophages to these injury parameters using clodronate liposomes, also demonstrating that AQP3 knockout macrophages exhibited a reduced increase in TNF α and alpha-SMA and an increased GSH/GSSG ratio. The effects seemed to be mediated by reactive oxygen species (ROS)-mediated activation of NF κ B, which was reduced in AQP3 knockout cells. Wild-type pro-inflammatory M1 macrophages increased oxidative stress and hydrogen peroxide secretion, the latter in an AQP3-, ROS- and NF κ B-dependent manner. AQP3 knockout macrophages were also less effective at activating hepatic stellate cells to express alpha-SMA or cause increased intracellular hydrogen peroxide levels and reduced GSH/GSSG ratios than wild-type macrophages. AQP3 knockout mice also showed less carbon tetrachloride-induced liver fibrosis and damage than the wild-type animals. Finally, these authors developed monoclonal antibodies to the extracellular domain of AQP3 and showed their ability to inhibit oxidative stress, NF κ B activation and TNF α and iNOS expression in, as well as hydrogen peroxide secretion from, macrophages. One of these monoclonal antibodies was then tested and shown to reduce carbon tetrachloride-induced liver fibrosis and damage in wild-type mice. The data are interesting and novel, and the development of this novel tool for inhibiting AQP3 activity is potentially highly significant. Some concerns arise, however, as detailed below.

Major points:

- (1) The authors have demonstrated that the monoclonal antibody inhibits hydrogen peroxide transport by AQP3. However, they provide no data on the effect of this antibody on AQP3's transport of water or glycerol. What is the effect of the monoclonal antibody on transport of water and glycerol through AQP3?
- (2) The authors observe some binding of the monoclonal antibody to AQP3 knockout cells (Figure 5c). To ensure that the monoclonal antibody is acting *in vivo* through its inhibition of AQP3 the authors should perform the carbon tetrachloride-induced liver injury model in AQP3 knockout mice treated with the monoclonal antibody versus IgG and show that there is no effect on the (lesser) liver damage observed in the AQP3 knockout mice. This experiment is also important since it is possible that AQP3 knockout mice have other compensatory changes resulting from loss of the AQP3 gene, which could influence their response to liver injury.
- (3) The data show that macrophages in the liver express AQP3, although it is not clear whether these macrophages are resting, M1 or M2. Indeed, the authors mention M2 macrophages but provided no data concerning them. Do M2 macrophages express AQP3 and if so, how does the expression compare to that in M1 or resting macrophages? How do AQP3 levels compare between resting and M1 macrophages?
- (4) According to the figure legends student t-tests were used for essentially all of the figures. Such tests are not appropriate for most, if not all, of the figures. A biostatistician should be consulted to ensure appropriate statistical analysis.
- (5) Western blotting results shown in Figures 2 and 5 should be quantified and cumulative results from at least 3 "n" illustrated and statistically analyzed.
- (6) AQP3 knockout does not seem to completely prevent liver damage, as seen by the increase in AST and ALT in Supplemental Figure 4, which should be discussed.

Minor points

- (1) In the abstract the AQP3 transport activity towards glycerol (and other small molecules) should be mentioned.
- (2) Was a single transcript verified for qPCR with SYBR Green?
- (3) The labeling for Figure 3e is not clear. Does the top x axis indicate how macrophages were treated and the bottom how the hepatic stellate cells were treated prior to co-culture? Were the indicated agents washed out before co-culture? In Figure 3h why was there no comparison with knockout cells treated with N-acetyl cysteine?
- (4) For Figure 4 were phospho-histone H2A γ -positive cells counted in a blinded fashion?
- (5) In Supplemental Figure 2d and e, what do the dark bars (wild-type?) and open bars (AQP3 knockout?) represent?

Reviewer #2:

Remarks to the Author:

Hara-Chikuma et al. investigated the role of AQP3 in CCl₄-induced liver injury and fibrosis. The study demonstrated global AQP3 knockout mice were resistant to CCl₄-induced liver injury and fibrosis. Then, the study found that AQP3 expression in macrophages plays a role in HSC activation. This was confirmed using co-culture experiments. Lastly, the team generated a new neutralizing Ab for AQP3, and determined that this Ab inhibited CCl₄-induced liver injury and fibrosis. Overall, the experimental design is reasonable, and the data provided are clear. However, to determine the universal anti-fibrotic effect of AQP3 inhibition, the data from two or three liver fibrosis should be provided. Moreover, the specific role of AQP3 in macrophages should be examined using in vivo conditional knockout or BM chimeric mouse model. Furthermore, liver macrophages (Kupffer cells) should be used for in vitro experiments rather than BM-derived macrophages. These points reduced the enthusiasm for this nice paper.

Specific comments:

1. While Figure 1a and b showed the expression of AQP3 in naïve mice, what is the expression of AQP3 in injured and fibrotic livers? In addition to the analysis in the animal model, what is the expression of AQP3 in human fibrotic livers?
2. In Figure 3, the study examined the inhibitory effect of AQP3^{-/-} in M1 macrophage polarization. Does AQP3 treatment polarize to M2 macrophages? Specifically, in vitro macrophage experiments should use liver macrophages, instead of BM-derived macrophages, throughout the study.
3. In Figure 3, co-culture experiments using macrophages and HSCs have been done. This phenotype should be demonstrated using in vivo models, for example, BM chimeric mice or conditional macrophage knockout mice.
4. P.8, l.27-, the text used "liver failure". This statement is incorrect. The study did not assess liver functions. Necrosis and leukocyte infiltration do not indicate "liver failure".
5. Quantification of Sirius red staining is missing in Figure 6e.
6. The effect of AQP3 deficiency and the therapeutic effect of mAb against AQP3 should be examined in the second liver fibrosis model, such as cholestasis- or NASH-induced liver fibrosis model.

Reviewer #1 (Remarks to the Author)

In this manuscript Hara-Chikuma et al. examine the role of aquaporin-3 (AQP3) and its transport of hydrogen peroxide in liver injury and fibrosis. Using AQP3 knockout mice with appropriate strain background controls, these authors show first that in wild-type mice AQP3 was expressed in macrophages and hepatic stellate cells in the liver (with low expression in hepatocytes) and that carbon tetrachloride induced a lesser increase in pro-inflammatory cytokines (TNF α and Ccl2), α -SMA (a marker of myofibroblasts) and oxidative stress, as well as a reduction in the ratio of reduced to oxidized glutathione (GSH/GSSG), despite having similar infiltration of macrophages into the liver. In wild-type mice they then showed the importance of macrophages to these injury parameters using clodronate liposomes, also demonstrating that AQP3 knockout macrophages exhibited a reduced increase in TNF α and α -SMA and an increased GSH/GSSG ratio. The effects seemed to be mediated by reactive oxygen species (ROS)-mediated activation of NF κ B, which was reduced in AQP3 knockout cells. Wild-type pro-inflammatory M1 macrophages increased oxidative stress and hydrogen peroxide secretion, the latter in an AQP3-, ROS- and NF κ B-dependent manner. AQP3 knockout macrophages were also less effective at activating hepatic stellate cells to express α -SMA or cause increased intracellular hydrogen peroxide levels and reduced GSH/GSSG ratios than wild-type macrophages. AQP3 knockout mice also showed less carbon tetrachloride-induced liver fibrosis and damage than the wild-type animals. Finally, these authors developed monoclonal antibodies to the extracellular domain of AQP3 and showed their ability to inhibit oxidative stress, NF κ B activation and TNF α and iNOS expression in, as well as hydrogen peroxide secretion from, macrophages. One of these monoclonal antibodies was then tested and shown to reduce carbon tetrachloride-induced liver fibrosis and damage in wild-type mice. The data are interesting and novel, and the development of this novel tool for inhibiting AQP3 activity is potentially highly significant. Some concerns arise, however, as detailed below.

Major points:

(1) The authors have demonstrated that the monoclonal antibody inhibits hydrogen peroxide transport by AQP3. However, they provide no data on the effect of this antibody on AQP3's transport of water or glycerol. What is the effect of the monoclonal antibody on transport of water and glycerol through AQP3?

RESPONSE:

As suggested, we have now investigated the effect of anti-AQP3 mAb on glycerol and

water transport. We found that the anti-AQP3 mAb fully inhibited glycerol uptake in AQP3 overexpressing cells to the level of that in non-AQP3-expressing control cells in a concentration-dependent manner. The new results have been added to Fig. 5c and Supplementary Fig. 4f-g.

The anti-AQP3 mAb also significantly, though partially inhibited osmotically induced water transport in AQP3 overexpressing cells (Supplementary Fig. 5h). The anti-AQP3 mAb may thus be more effective against H₂O₂ and glycerol transport than water transport, perhaps because of different steric restrictions in the AQP3 pore region. The new data (Supplementary Fig. 5h) and discussion have been included in the revised manuscript (page 10, line 29-).

(2) The authors observe some binding of the monoclonal antibody to AQP3 knockout cells (Figure 5c). To ensure that the monoclonal antibody is acting *in vivo* through its inhibition of AQP3 the authors should perform the carbon tetrachloride-induced liver injury model in AQP3 knockout mice treated with the monoclonal antibody versus IgG and show that there is no effect on the (lesser) liver damage observed in the AQP3 knockout mice. This experiment is also important since it is possible that AQP3 knockout mice have other compensatory changes resulting from loss of the AQP3 gene, which could influence their response to liver injury.

RESPONSE:

As suggested, we have done the CCl₄-induced liver injury model in AQP3 knockout mice treated with anti-AQP3 mAb or control IgG. We found that the anti-AQP3 mAb did not affect the liver damage in AQP3 knockout mice, and no compensatory changes were seen in AQP3 knockout mice with the mAb. These results are provided in Supplemental fig. 7b and 7c, and the corresponding text (page 11, line 26-).

(3) The data show that macrophages in the liver express AQP3, although it is not clear whether these macrophages are resting, M1 or M2. Indeed, the authors mention M2 macrophages but provided no data concerning them. Do M2 macrophages express AQP3 and if so, how does the expression compare to that in M1 or resting macrophages? How do AQP3 levels compare between resting and M1 macrophages?

RESPONSE:

We have now studied AQP3 expression by real-time RT-PCR in resting (control), M1 (LPS and IFN- γ treated), and M2 (IL-4 treated) macrophages. We found that AQP3 expression was greater in M2 macrophages compared to resting and M1 macrophages.

Of note, we confirmed that differentiation of macrophages into the M2 subtype by IL-4

stimulation, quantified by arginase 1 (ARG1) as an M2 marker, was comparable in WT and AQP3 knockout macrophages, while differentiation into the M1 subtype was reduced in AQP3 knockout cells, as shown in the original Fig. 3a. The new data have been added (Supplementary Fig. 3a-c) and in the corresponding text (page 8, line 11-15).

(4) According to the figure legends student t-tests were used for essentially all of the figures. Such tests are not appropriate for most, if not all, of the figures. A biostatistician should be consulted to ensure appropriate statistical analysis.

RESPONSE:

We have now performed the appropriate statistical analyses (using GraphPad Prism 8 program), with information included in each figure legend in the revised manuscript.

(5) Western blotting results shown in Figures 2 and 5 should be quantified and cumulative results from at least 3 “n” illustrated and statistically analyzed.

RESPONSE:

We have performed Western blots on 3 or more different samples from at least two independent experiments, with representative immunoblots provided. As suggested, we have now quantified the ratio of phospho-p65 to p65, as provided Supplementary Fig. 2f and 5i.

(6) AQP3 knockout does not seem to completely prevent liver damage, as seen by the increase in AST and ALT in Supplemental Figure 4, which should be discussed.

RESPONSE:

As suggested, we have mentioned this point in the revised text (page 9, line 22).

Minor points

(1) In the abstract the AQP3 transport activity towards glycerol (and other small molecules) should be mentioned.

RESPONSE:

As mentioned above, we found inhibition of AQP3-facilitated glycerol transport. This result is revised Fig. 5C as well as the revised abstract.

(2) Was a single transcript verified for qPCR with SYBR Green?

RESPONSE:

Yes, a single transcript was verified for qPCR with SYBR green.

(3) The labeling for Figure 3e is not clear. Does the top x axis indicate how macrophages were treated and the bottom how the hepatic stellate cells were treated prior to co-culture? Were the indicated agents washed out before co-culture? In Figure 3h why was there no comparison with knockout cells treated with N-acetyl cysteine?

RESPONSE:

To clarify these issues, the figure format has been modified and additional information and explanation have been added in the revised text and legend to Fig. 3e. As for Fig 3h, data for AQP3 knockout cells treated with N-acetyl cystein (NAC) are included to the revised Figure 3h.

(4) For Figure 4 were phospho-histone H2A gamma-positive cells counted in a blinded fashion?

RESPONSE:

Yes, the brown color-H2A gamma positive cells were counted by Tissue Quest in a blinded manner. This information has been added to the method (page 17, line 10).

(5) In Supplemental Figure 2d and e, what do the dark bars (wild-type?) and open bars (AQP3 knockout?) represent?

RESPONSE:

We added the information for the bars (dark: WT, open: AQP3 knockout) in the revised Fig 2e (original sup Fig 2e).

Thank you for pointing out.

Reviewer #2 (Remarks to the Author):

Hara-Chikuma et al. investigated the role of AQP3 in CCl₄-induced liver injury and fibrosis. The study demonstrated global AQP3 knockout mice were resistant to CCl₄-induced liver injury and fibrosis. Then, the study found that AQP3 expression in macrophages plays a role in HSC activation. This was confirmed using co-culture experiments. Lastly, the team generated a new neutralizing Ab for AQP3, and determined that this Ab inhibited CCl₄-induced liver injury and fibrosis. Overall, the experimental design is reasonable, and the data provided are clear. However, to determine the universal anti-fibrotic effect of AQP3 inhibition, the data from two or three liver fibrosis should be provided. Moreover, the specific role of AQP3 in macrophages should be examined using in vivo conditional knockout or BM chimeric mouse model. Furthermore, liver macrophages (Kupffer cells) should be used for in vitro experiments rather than BM-derived macrophages. These points reduced the enthusiasm for this nice paper.

RESPONSE:

We appreciate these helpful comments. We have now done two additional liver injury models. In an azoxymethane (AOM)-induced liver injury model we found that AQP3 knockout mice had significantly reduced acute liver injury as well as chronic liver injury and fibrosis. Further, we confirmed that the anti-AQP3 mAb suppressed AOM-induced acute liver injury in WT mice (Sup Fig. 8a-g). In addition, we have done the CCl₄-induced liver injury model using the bone marrow chimeric mouse model (Fig. 1g). Finally, we have added new data using liver macrophages (Fig. 2h,i). Details follow.

Specific comments:

1. While Figure 1a and b showed the expression of AQP3 in naïve mice, what is the expression of AQP3 in injured and fibrotic livers? In addition to the analysis in the animal model, what is the expression of AQP3 in human fibrotic livers?

RESPONSE:

We studied AQP3 expression in liver tissues in CCl₄ chronic model (vs. vehicle treatment) by immunostaining with anti-AQP3 and anti-F4/80 (macrophage marker). Immunostaining showed that AQP3 was expressed in F4/80⁺ macrophages that were increased by CCl₄ treatment. These data have been added in revised Supplementary Fig 4c and the accompanying text (page 9, line 27-).

We have done AQP3 immunostaining in a human liver tissue array that included 14 normal liver tissues and 21 tissues with chronic active hepatitis (from US Biomac, #LV1201). The new immunostaining micrographs have been added in revised

Supplementary Fig. 1a) and the corresponding text (page 5, line 5-).

2. In Figure 3, the study examined the inhibitory effect of AQP3^{-/-} in M1 macrophage polarization. Does AQP3 treatment polarize to M2 macrophages? Specifically, in vitro macrophage experiments should use liver macrophages, instead of BM-derived macrophages, throughout the study.

RESPONSE:

We confirmed that differentiation of macrophages into the M2 subtype by sustained IL-4 stimulation, quantified by arginase 1 (ARG1) as M2 marker, was comparable in WT and AQP3 knockout cells, while differentiation into the M1 subtype was reduced in AQP3 knockout cells, as shown in the original Fig. 3a. The new data have been added (Supplementary Fig. 3a) and the corresponding text (page 8, line 11-).

The original data for Figures 1d, 1e, 2f, 2g, 3e, and 3f were done with liver macrophages. In addition, we performed key experiments using liver macrophages, which were added as Figures 2h and 2i and the corresponding text (page 7, line 11-).

3. In Figure 3, co-culture experiments using macrophages and HSCs have been done. This phenotype should be demonstrated using in vivo models, for example, BM chimeric mice or conditional macrophage knockout mice.

RESPONSE:

We have now done the CCl₄-induced liver injury model using the bone marrow chimeric mouse model, and verified the involvement of AQP3 expression in hematopoietic cells including macrophages. The new data have been added to Fig. 1g and Supplementary Fig. 1e, and corresponding text (page 6, line 3-).

4. P.8, 1.27-, the text used “liver failure”. This statement is incorrect. The study did not assess liver functions. Necrosis and leukocyte infiltration do not indicate “liver failure”.

RESPONSE:

The text has been appropriately amended throughout the manuscript.

5. Quantification of Sirius red staining is missing in Figure 6e.

RESPONSE:

The original Supplementary Fig 7 included quantification of Sirius red staining corresponding to Fig 6e. This figure has moved to Fig 6e in the revised text.

6. The effect of AQP3 deficiency and the therapeutic effect of mAb against AQP3 should be

examined in the second liver fibrosis model, such as cholestasis- or NASH-induced liver fibrosis model.

RESPONSE:

We have now performed liver injury models involving azoxymethane (AOM) injection. We found that AQP3 knockout mice had reduced AOM-induced acute liver injury (Supplementary Fig. 8d), as well as chronic liver injury and fibrosis (Supplementary Fig. 8a-c). Further, we confirmed that the anti-AQP3 mAb reduced AOM-induced acute liver injury in WT mice (Supplementary Fig. 8e-g). The new data have been added to Supplementary Fig.8a-g and references (reference # 45-47), and text (page 11, line 29-).

Reviewers' Comments:

Reviewer #1:

Remarks to the Author:

In this revised manuscript Hara-Chikuma et al. examine the role of aquaporin-3 (AQP3) and its transport of hydrogen peroxide in liver injury and fibrosis. The data are interesting and novel, and the authors have responded well to previous critiques, supplying a good deal of new data to strengthen the manuscript. However, the statistical analyses performed are still often incorrect (see below). Nevertheless, the development of this novel tool for inhibiting AQP3 activity is potentially highly significant, and the manuscript should be of great interest to a large number of scientists.

Major point:

(1) According to the figure legends student t-tests were still used for a majority of the figures. Such tests are not appropriate for figures in which more than two groups are compared. Thus, Figure 1c-f, Figure 2a-c, f-j, Figure 3a-c, Figure 4b-g, Figure 5e and f, and several figures in the Supplemental Materials require statistical analyses other than Student's t-tests, ideally a two-way ANOVA (or three-way in some cases). It seems likely that two-way analysis will show an interaction between genotype and treatment, in which case ANOVA can be used to determine significance between the groups. As stated previously, the authors are urged to consult a biostatistician to ensure appropriate statistical analysis.

Minor points:

- (1) The third and fourth sentence of the abstract seem redundant.
- (2) On page 8 the authors should indicate where the data are shown that are described in the last two sentences.

Reviewer #2:

Remarks to the Author:

The authors responded most of the comments from this reviewer. However, there are still have a few concerns as described below.

In response to this reviewer's comment, authors assessed AQP3 expression in human livers. Although Supp Fig 1a Healthy (Male, 47y) showed nice AQP3 expression in non-parenchymal cells, but not hepatocytes, but the staining quality of other sections is poor, and it is impossible to tell AQP3 expression is upregulated in liver macrophages in chronic inflamed livers.

Authors additionally performed BM-chimeric study and have shown in Figure 1g. However, Figure 1g indicated that AQP3 in recipient cells are relatively more important than that in donor BM cells. The effect of AQP3^{-/-} in bone-marrow cells was also seen, but CCL2 is not significantly reduced in WT recipients with KO donors. The authors suggested HSC may play a role. It might be true. However, the result from BM-chimeric mice is inconsistent with the co-culture experiment shown in Figure 3f. What is the role of AQP3 in HSCs? This point has not been investigated.

Because AQP3^{-/-} did not show the effect of M2 polarization, AQP3^{-/-} effect on M1 polarization might not be a right interpretation. AQP3 might not affect M1 and M2 polarization. Rather, AQP3^{-/-} inhibited LPS/IFN γ -induced inflammatory response.

This reviewer has never seen the AOM model as a mouse liver fibrosis. It is unclear whether this is a human relevant liver fibrosis preclinical model.

Reviewers' comments:

Reviewer #1 (Remarks to the Author):

In this revised manuscript Hara-Chikuma et al. examine the role of aquaporin-3 (AQP3) and its transport of hydrogen peroxide in liver injury and fibrosis. The data are interesting and novel, and the authors have responded well to previous critiques, supplying a good deal of new data to strengthen the manuscript. However, the statistical analyses performed are still often incorrect (see below). Nevertheless, the development of this novel tool for inhibiting AQP3 activity is potentially highly significant, and the manuscript should be of great interest to a large number of scientists.

Major point:

(1) According to the figure legends student t-tests were still used for a majority of the figures. Such tests are not appropriate for figures in which more than two groups are compared. Thus, Figure 1c-f, Figure 2a-c, f-j, Figure 3a-c, Figure 4b-g, Figure 5e and f, and several figures in the Supplemental Materials require statistical analyses other than Student's t-tests, ideally a two-way ANOVA (or three-way in some cases). It seems likely that two-way analysis will show an interaction between genotype and treatment, in which case ANOVA can be used to determine significance between the groups. As stated previously, the authors are urged to consult a biostatistician to ensure appropriate statistical analysis.

RESPONSE:

We really appreciate pointing out our mistakes. We have now done the appropriate statistical analyses (using GraphPad Prism 8) following consultation with a statistician. The new statistical information has been added in each figure legend in the revised manuscript.

Minor points:

(1) The third and fourth sentence of the abstract seem redundant.

RESPONSE:

As suggested, we have reworded the third and fourth sentences of the abstract.

(2) On page 8 the authors should indicate where the data are shown that are described in the last two sentences.

RESPONSE:

We now mention directly where the data are shown related to the two sentences in question.

Reviewer #2 (Remarks to the Author):

1. The authors responded most of the comments from this reviewer. However, there are still have a few concerns as described below.

In response to this reviewer's comment, authors assessed AQP3 expression in human livers. Although Supp Fig 1a Healthy (Male, 47y) showed nice AQP3 expression in non-parenchymal cells, but not hepatocytes, but the staining quality of other sections is poor, and it is impossible to tell AQP3 expression is upregulated in liver macrophages in chronic inflamed livers.

RESPONSE:

We now obtained a new liver disease array from US Biomax (#LC805a) and performed additional AQP3 immunostaining to increase the sample number and improve the staining quality in normal and cirrhosis liver. To quantify AQP3 expression in parenchymal cells, three independent experts blinded to the pathological information (total 23 normal liver and 63 hepatic cirrhosis) scored AQP3 intensity. The new results have been added to Supp Fig. 1a and text has been reworded (page 5, line 5-8).

2. Authors additionally performed BM-chimeric study and have shown in Figure 1g. However, Figure 1g indicated that AQP3 in recipient cells are relatively more important than that in donor BM cells. The effect of AQP3^{-/-} in bone-marrow cells was also seen, but CCL2 is not significantly reduced in WT recipients with KO donors. The authors suggested HSC may play a role. It might be true. However, the result from BM-chimeric mice is inconsistent with the co-culture experiment shown in Figure 3f. What is the role of AQP3 in HSCs? This point has not been investigated.

RESPONSE:

Thank you for this comment. To determine the role of AQP3 in HSC activation, isolated HSCs from wild-type and AQP3 knockout were activated by TNF- α . We found that mRNA expression of alpha-SMA and TGF- β 1 after TNF- α induction were similar in wild-type and AQP3 knockout HSCs (Sup Fig. 3d). After consideration of this result and the data from co-culture of HSC and macrophages (Fig. 3f and Sup Fig. 3f), we conclude that macrophages rather than HSCs are involved in the reduced liver injury in AQP3 knockout mice.

We agree with the reviewer that the results with bone marrow chimeric mice shown in Fig. 1g suggest that AQP3 in recipient cells might be relatively more important than that in donor BM cells, and thus cannot exclude the possibility that AQP3-expressing HSCs plays a role. On the other hand, in our method of generating

bone marrow chimeric mice, the recipient cells contain resident hepatic macrophages (Kupffer cells). Therefore, the resident hepatic macrophages might have some effect on recipient mice. We have added the new results to Sup Fig. 3d with explanation added in the text (page 6, line 11-13; page 8, line 23-28).

3. Because AQP3^{-/-} did not show the effect of M2 polarization, AQP3^{-/-} effect on M1 polarization might not be a right interpretation. AQP3 might not affect M1 and M2 polarization. Rather, AQP3^{-/-} inhibited LPS/IFN γ -induced inflammatory response.

RESPONSE:

After careful consideration, we agree with the reviewer's point and have reworded the corresponding text (page 8, line 9-11, 15-17; page 9, line 15; page 11, line 17).

4. This reviewer has never seen the AOM model as a mouse liver fibrosis. It is unclear whether this is a human relevant liver fibrosis preclinical model.

RESPONSE:

Published studies reported that AOM injection induced acute liver injury and chronic liver fibrosis, as relevant to human disease (refs. 1-7 below), which are now cited.

1. Matkowskyj KA, Marrero JA, Carroll RE, Danilkovich AV, Green RM, Benya RV. Azoxymethane-induced fulminant hepatic failure in C57BL/6J mice: characterization of a new animal model. *Am J Physiol* **277**, G455-462 (1999).
2. Nishihara T, *et al.* Adiponectin deficiency enhances colorectal carcinogenesis and liver tumor formation induced by azoxymethane in mice. *World J Gastroenterol* **14**, 6473-6480 (2008).
3. Chastre A, Belanger M, Beauchesne E, Nguyen BN, Desjardins P, Butterworth RF. Inflammatory cascades driven by tumor necrosis factor- α play a major role in the progression of acute liver failure and its neurological complications. *PLoS One* **7**, e49670 (2012).
4. Rachakonda V, *et al.* M1 muscarinic receptor deficiency attenuates azoxymethane-induced chronic liver injury in mice. *Sci Rep* **5**, 14110 (2015).
5. Khurana S, *et al.* Effects of modulating M3 muscarinic receptor activity on azoxymethane-induced liver injury in mice. *Biochem Pharmacol* **86**, 329-338

(2013).

6. Bemeur C, Desjardins P, Butterworth RF. Antioxidant and anti-inflammatory effects of mild hypothermia in the attenuation of liver injury due to azoxymethane toxicity in the mouse. *Metab Brain Dis* **25**, 23-29 (2010).
7. Bemeur C, Qu H, Desjardins P, Butterworth RF. IL-1 or TNF receptor gene deletion delays onset of encephalopathy and attenuates brain edema in experimental acute liver failure. *Neurochem Int* **56**, 213-215 (2010).

Reviewers' Comments:

Reviewer #1:

Remarks to the Author:

The authors have responded well to previous critiques (although I would note that in Supplemental Figure 1 "Healthy" is the correct spelling).

Reviewer #2:

Remarks to the Author:

To answer this reviewer's previous comment #1, Authors increased the number of human samples and assessed AQP3 expression in liver tissues. More obviously, AQP3 is predominantly expressed in hepatocytes, but not expressed in liver macrophages. This data did not support the authors' hypothesis and the other mouse data (no or less expression of AQP3 in mouse hepatocytes).

To answer this reviewer's previous comment #2, authors performed an additional in vitro experiment using HSCs treated with TNF α , and found there is no difference between WT and KO cells. The data might be true. However, TNF α is known to not activate HSCs, rather contribute to the HSC survival (PMID: 23553591). Authors might have missed very important observations. Moreover, authors claimed resident hepatic macrophages might have affected the phenotype of BM chimeric mice. If this is the case, the authors' approach using BM-chimeric mice was not the right approach. Thus, the response of authors to comment #2 is unsatisfactory. Figure 1g obviously indicated that recipient cells, HSCs (and/or hepatocytes), are relatively more important than BM-derived cells in AQP3-mediated liver injury and fibrosis.

In response to comment #4, authors referred several papers. However, those papers were not published in the well-recognized liver research journals. Am J Physiol (the paper from Matlowskyj et al) is a historical journal, but this paper has been published in 20 years ago. Thus, the AOM model is not a modern liver fibrosis model.

RESPONSE to REVIEWER #2

1. To answer this reviewer's previous comment #1, Authors increased the number of human samples and assessed AQP3 expression in liver tissues. More obviously, AQP3 is predominantly expressed in hepatocytes, but not expressed in liver macrophages. This data did not support the authors' hypothesis and the other mouse data (no or less expression of AQP3 in mouse hepatocytes).

RESPONSE:

To clarify AQP3 expression in normal liver, we purchased commercially available frozen liver tissue from healthy subjects and performed the double immunofluorescence staining with anti-AQP3 and anti-CD68 (human macrophage marker). AQP3 was expressed mainly in macrophages in normal human liver (Supplementary Fig 1a), in agreement with findings in control mouse liver (shown in Fig. 1a and 1b). As the reviewer pointed out, AQP3 staining appears to be increased in hepatocytes of injured liver, including in human cirrhosis (Supplementary Fig 1b) and CCl₄-treated mice (Supplementary Fig 4c). From these results we conclude that AQP3 is expressed mainly in macrophages and hepatic stellate cells in normal liver, and its expression is increased in hepatocytes with injury or inflammation.

The new results have been added to Supplementary Fig 1a and in the accompanying text (page 5, lines 5-7) and supplementary methods. We thank the reviewer for bringing this to our attention.

2. To answer this reviewer's previous comment #2, authors performed an additional *in vitro* experiment using HSCs treated with TNF α , and found there is no difference between WT and KO cells. The data might be true. However, TNF α is known to not activate HSCs, rather contribute to the HSC survival (PMID: 23553591). Authors might have missed very important observations. Moreover, authors claimed resident hepatic macrophages might have affected the phenotype of BM chimeric mice. If this is the case, the author's approach using BM-chimeric mice was not the right approach. Thus, the response of authors to comment #2 is unsatisfactory. Figure 1g obviously indicated that recipient cells, HSCs (and/or hepatocytes), are relatively more important than BM-derived cells in AQP3-mediated liver injury and fibrosis.

RESPONSE:

We agree that Figure 1g supports some involvement of HSCs in AQP3-mediated liver injury. We note, however, that the *in vitro* co-culture experiment of macrophages

with HSCs showed that AQP3 expression in macrophages is relatively more important than AQP3 expression in HSCs (shown in Figure 3f and Supplementary Figure 3f). A series of experiments with macrophage depletion or transfer directly demonstrated the importance of AQP3 expression in macrophages during acute liver injury (Figures 2a-e). Moreover, low binding affinity of the anti-AQP3 mAb to HSCs and hepatocytes was found (new Supplementary Figure 5i). It follows that the efficacy of the anti-AQP3 mAb in blocking liver injury results from inhibition of macrophage AQP3.

We underscore that the major finding in this paper is the development of novel neutralizing anti-AQP3 mAb, and demonstration of its efficacy in preventing liver injury in several experimental animal models. Though further studies are required to elucidate the role of AQP3 in HSCs, we believe that our findings fully support the conclusion that anti-AQP3 mAb is effective in preventing liver injury through action on AQP3 expressing macrophages.

The new binding data for the anti-AQP3 mAb are shown in Supplementary Fig. 5i and corresponding text (page 11, lines 17-18).

3. In response to comment #4, authors referred several papers. However, those papers were not published in the well-recognized liver research journals. Am J Physiol (the paper from Matlowkyj et al) is a historical journal, but this paper has been published in 20 years ago. Thus, the AOM model is not a modern liver fibrosis model.

RESPONSE:

When we determined the role of AQP3 expression in colon carcinogenesis using well known AOM-induced colon cancer mouse model, we found that wild-type liver showed hepatitis-like symptoms with lots of tumors, but much less symptoms in AQP3 knockout liver. These preliminary results were our motivation to examine the role of AQP3 in liver injury and the effect of AQP3 antibody.

We acknowledge that the AOM injection model is not the commonly used model of liver injury. As suggested, we have now completed a full set of studies using the well-established liver injury model involving thioacetamide (TAA) injection. We found that AQP3 knockout suppressed TAA-induced liver injury, and that anti-AQP3 mAb administration reduced TAA-induced acute liver injury. These new experiments are reported in Supplemental Fig. 8a-e, and in the revised manuscript text (page 12, lines 14-31). We thank the reviewer for his/her insistence that we test this model, as the results nicely strengthen the central conclusions of our paper involving the importance of AQP3 in liver injury and the efficacy of the anti-AQP3 mAb.

Reviewers' Comments:

Reviewer #2:

Remarks to the Author:

The authors responded to this reviewer's previous comments. I still have a few comments.

The authors included a new staining for human liver samples in Supplementary Figure 1a. These pictures show nice overlapping of AQP3 with CD68 expressing macrophages. Although there are several AQP3 positive areas that are not macrophages, the staining has been improved a lot. However, this reviewer still disagrees the IHC shown in Supplementary Figure 1b. The staining in hepatocytes should be non-specific, which might be due to insufficient avidin, biotin blockade. Because Supplementary Figure 1b does not support overall hypothesis of this study, this reviewer highly recommends the authors as well as the editors to remove Supplementary Figure 1b from the manuscript.

This reviewer disagrees the response from the authors, AQP3 expression was increased in hepatocytes in CCl₄-treated mice (Supplementary Figure 4c). The co-staining picture shows the majority of AQP3 are co-stained with F4/80. However, there is no need to revise the manuscript because the authors did not claim this point in the manuscript.

RESPONSE to REVIEWER #2

The authors responded to this reviewer's previous comments. I still have a few comments. The authors included a new staining for human liver samples in Supplementary Figure 1a. These pictures show nice overlapping of AQP3 with CD68 expressing macrophages. Although there are several AQP3 positive areas that are not macrophages, the staining has been improved a lot. However, this reviewer still disagrees the IHC shown in Supplementary Figure 1b. The staining in hepatocytes should be non-specific, which might be due to insufficient avidin, biotin blockade. Because Supplementary Figure 1b does not support overall hypothesis of this study, this reviewer highly recommends the authors as well as the editors to remove Supplementary Figure 1b from the manuscript.

This reviewer disagrees the response from the authors, AQP3 expression was increased in hepatocytes in CCl4-treated mice (Supplementary Figure 4c). The co-staining picture shows the majority of AQP3 are co-stained with F4/80. However, there is no need to revise the manuscript because the authors did not claim this point in the manuscript.

RESPONSE:

We thank the reviewer for this suggestion. After careful consideration, we agree with the reviewer. As recommended, we have removed the IHC result in Supplementary Figure 1b and reworded the corresponding text to Supplementary Figures 1b and 4c (page 5, line 5-7, page 10, line 1-3).

We thank the reviewer for bringing this to our attention.